



# 1 Satellite Imagery and Products of the 16-17 February 2020 Saharan Air Layer
# 2 Dust Event over the Eastern Atlantic: Impacts of Water Vapor on Dust
# 3 Detection and Morphology

Lewis Grasso[1], Dan Bikos[1], Jorel Torres[1], John F. Dostalek[1], Ting-Chi Wu[1], John Forsythe[1],
Heather Q. Cronk[1], Curtis J. Seaman[1], Steven D. Miller[1], Emily Berndt[2], Harry G. Weinman[3], and
Kennard B. Kasper[4]
[1]Cooperative Institute for Research in the Atmosphere, Colorado State University, Fort Collins, CO
[2]NASA Marshall Space Flight Center, Short-term Prediction Research and Transition Center, Huntsville, AL
[3]NOAA/NWS Miami-South Florida Weather Forecast Office, Miami, FL
[4]NOAA/NWS Florida Keys Weather Forecast Office, Key West, FL
*Correspondence to:* Lewis D. Grasso (Lewis.Grasso@colostate.edu)
**Abstract.** On 16-17 February 2020, dust within a Saharan Air Layer (SAL) from western Africa moved over the
eastern Atlantic Ocean. Satellite imagery and products from ABI on GOES-16, VIIRS on NOAA-20, and CALIOP
on CALIPSO along with retrieved values of layer and total precipitable water (TPW) from MiRS and NUCAPS,
respectively, were used to identify dust within the SAL over the eastern Atlantic Ocean. Use of various satellite
imagery and products were also used to characterize the distribution of water vapor within the SAL. There was a
distinct pattern between dust detection and dust masking and values of precipitable water. Specifically, dust was
detected when values of layer or TPW were approximately 14 mm; in addition, dust was masked when values of layer
or TPW were approximately 28 mm. In other words, water vapor masked infrared dust detection if sufficient amounts
of water vapor existed in a column. Results herein provide observational support to two recent numerical studies that
concluded water vapor can mask infrared detection of airborne dust.

## 22 1 Introduction

For over forty-five years, satellite data has been used to detect airborne dust. Detection of dust has been explored with
the use of Low Earth Orbiting (LEO) sensors such as the (i) Moderate Resolution Imaging Spectroradiometer (King
et al. 1992), (ii) Cloud-Aerosol Lidar with Orthogonal Polarization (CALIOP; Winker et al. 2009), and (iii)
Temperature Humidity Infrared Radiometer and Image Dissector Camera System both onboard Nimbus-4 (Shenk and
Curran 1974). In addition, geostationary sensors such as the (i) Spinning Enhanced Visible and InfraRed Imager
(SEVIRI) onboard METEOSAT Second Generation (MSG) (Schmetz et al. 2002) and (ii) Advanced Baseline Imager
(ABI; Kalluri et al. 2018, Schmit et al. 2008) onboard GOES-16/17 have been used to explore dust. Platforms orbiting
the Earth allowed for many types of techniques to detect airborne dust.
Typically several types of procedures exist that use a variety of spectral bands to detect dust in the atmosphere of the
Earth. For example, Ashpole and Washington (2012), Knippertz and Todd (2010), Torres et al. (1998), Torres et al.
(2007), and Herman et al. (1997) used spectral bands in the ultraviolet for dust detection. In addition, techniques have
also been developed that required only spectral bands in the infrared (Lensky and Rosenfeld 2008; Chaboureau et al.





2007; Darmenov and Sokolik 2005; Ackerman 1997; Legrand et al. 1989; Shenk and Curran 1974). There also exist
dust detection algorithms that used a combination of spectral bands that detect both solar reflection and infrared energy
(Cho et al. 2013; Zhao et al. 2010; Hao and Qu 2007; Pierangelo et al. 2004; Miller 2003, Miller et al. 2017; Legrand
et al. 2001; Tanre and Legrand 1991; Ackerman 1989). Of the above studies, some have speculated about the
relationship between water vapor and dust detection.
An open question centers on the inter-dependence between water vapor and dust detection algorithms. Although each
of the above studies focused on examining which spectral bands may be used for dust detection, a few did raise the
question on possible effects, either advantageously or adversely, of water vapor on dust detection (Ashpole and
Washington 2012; Knippertz and Todd 2010; Chaboureau et al. 2007; Legrand et al. 2001; Tanre and Legrand 1991).
Interestingly, Pierangelo et al. (2004), pointed out that dust interfered with temperature and water vapor retrievals.
Recent work, however, directly addressed the impact water vapor may have on dust detection.
This paper endeavors to extend a few recent studies. That is, the statement of the problem in this manuscript is that
two recent studies explored the use of numerical modeling to support an hypothesis that water vapor has the ability to
mask dust (Banks et al. 2019; Miller et al. 2019). This manuscript examines an observational case of a Saharan Air
Layer (SAL; Prospero and Carlson, 1972; Adams et al., 2012; Dunion and Velden, 2004; Kuciauskas et al. 2018) dust
event over the eastern Atlantic Ocean.  Results of the 16-17 February 2020 observational study serve to support Banks
et al. (2019) and Miller et al. (2019) by showing that reduced values of water vapor allowed dust associated with the
SAL to be both detected and tracked over the eastern Atlantic Ocean. Observational datasets include 1) a simple
difference in infrared imagery, 2) a microwave retrieval of layer precipitable water known as the Advected Layer
Precipitable Water (ALPW) product (Forsythe et al. 2015; LeRoy et al. 2016), and 3) an infrared retrieval of total
precipitable water (TPW; Gambacorta and Barnet 2013; Gambacorta 2013), both of which addressed water vapor in
the environment of the SAL  dust.
Organization of the manuscript is as follows: A detailed discussion of sources of satellite and retrieved data is found
in Section 2. An in depth examination and interpretation of the SAL dust plume as revealed by remote imagery is the
focus of Section 3. Assimilation of dust is a relatively new effort; as such, a brief overview of recent efforts of dust
assimilation are contained in Section 4. National Weather Service forecasters provide pertinent forecasting issues
associated with SAL dust events along with potential impacts of SAL dust over South Florida in Section 5, which is
entitled, "Forecaster Perspective". Finally, the summary and conclusions are provided in Section 6.
**2 Satellite Data**
Satellite data from three sources and retrieved precipitable water from two source were used for this study. Satellite
data was acquired from the (1) Advanced Baseline Imager (ABI) onboard the Geostationary Operational
Environmental Satellite (GOES)-16, (2) Visible Infrared Imaging Radiometer Suite (VIIRS) onboard the National
Oceanic and Atmospheric Administration (NOAA)-20, and (3) Cloud-Aerosol Lidar with Orthogonal Polarization



(CALIOP), which is onboard the Cloud–Aerosol Lidar Infrared Pathfinder Satellite Observations (CALIPSO).
Retrieved precipitable water was acquired from (1) the NOAA Unique Combined Atmospheric Processing System
(NUCAPS) and (2) the Microwave integrated Retrieval System (MiRS). Although this manuscript focuses on a SAL
dust plume that moved from western Africa to the eastern Atlantic Ocean, the lack of a blue band (~ 0.47 μm) on
SEVIRI, which is onboard MSG-11, whose sub-point is the intersection of the prime meridian and the equator,
prevents the generation of Geo/True-Color imagery. As a result, discussions about the airborne dust will utilize the
above sources. A brief discussion of each data source will be discussed presently; additional information may be found
in the included references.
On 19 November 2016, GOES-R was launched from Kennedy Space Center, Cape Canaveral, Florida. After reaching
a position at 89.5 W and undergoing a check-out period, the satellite assumed an operational identification of GOES-
16 and currently resides at 75.2 W. Imagery from ABI, one of the primary sensors on GOES-16, was used for this
study. Unlike previous GOES imagers, ABI collects imagery at sixteen different spectral bands with a nadir
instantaneous geometric field of view of 0.5, 1.0, and 2.0 km (Kalluri et al. 2018; Schmit et al. 2008; Goodman et al.
2012). There exists several applications for data from ABI, some of which  are the following: GeoColor imagery
(Miller et al. 2016, 2017), cloud properties (Heidinger et al. 2015), land/ocean surfaces, the cryosphere, atmospheric
soundings, and atmospheric aerosol (Schmit et al. 2017, 2018). Additional information about GOES-16 and other
satellites in the GOES-R series may be found in Goodman et al. (2019).
VIIRS was first placed on the Suomi National Polar-orbiting Partnership (S-NPP) platform, which was launched in
28 October 2011. S-NPP served as a demonstration, as opposed to an operational, satellite. Due to the success of S-
NPP, VIIRS was placed onboard the Joint Polar Satellite System (JPSS) (Goldberg et al., 2013). On November 2017,
JPSS-1 was launched; the satellite assumed the operational identification of NOAA-20 on 18 November 2018. Both
S-NPP and NOAA-20 are in the same orbital plane and are separated by approximately one-half orbit, allowing for
two VIIRS images every ~50 minutes. VIIRS allows for imaging of footprints at both 750 m for M-bands and 375 m
for I-bands.  VIIRS swath widths are approximately 3,000 km; further, VIIRS contains a Day Night Band, which has
the ability to capture several features at night due to reflected moon light and surface light sources. A detailed list of
applications and capabilities of VIIRS are discussed in Hillger et al. (2013, 2014) and Miller et al. (2012, 2013).
On 28 April 2006, CALIPSO was launched and positioned in the A-Train constellation of low Earth orbiting satellites.
CALIOP (Winker et al., 2009), is the main sensor onboard CALIPSO. A component of CALIOP is a lidar, which
emits packets of 110 mJ of energy at a frequency of 20.25 Hz downward to the surface of the Earth. In addition,
backscattered energy is detected at 532 nm, polarized 532 nm, and 1064 nm by three detectors; additional details are
described in Hunt et al. (2009). CALIOP acquires data that is used to produce the Vertical Feature Mask (VFM). VFM
is a vertically oriented plane within which certain atmospheric constituents, if present, are identified. Some of the
identifiable constituents are, but not limited to, clear sky, clouds, and aerosols (Liu et al., 2005). CALIOP differs from
ABI and VIIRS; specifically, both ABI and VIIRS are passive sensors while CALIOP is an active sensor.



Retrieved atmospheric soundings of temperature and water vapor were acquired from the NOAA Unique Combined
Atmospheric Processing System (NUCAPS; Gambacorta and Barnet 2013; Gambacorta 2013), which is a NOAA
operational algorithm for hyperspectral infrared retrievals. A modular algorithm design allows NUCAPS to be applied
to hyperspectral infrared sounders on multiple satellite platforms. In the case of the S-NPP/NOAA-20 series of
satellites, NUCAPS uses input from the Cross-track Infrared Sounder and the Advanced Technology Microwave
Sounder sensors. NUCAPS also uses cloud-cleared radiances and an iterative regularized least squares minimization
algorithm to produce vertical profiles of temperature and water vapor from microwave and infrared radiances. Thirty
retrievals are performed across a 2200 km swath, with footprint sizes ranging from ~50 km at nadir to 70 km x 134
km at the edge. Retrieved profiles are mapped onto 100 vertical levels between 1100 hPa and 0.016 hPa. Examples
of applications include the "Cold-air aloft" aviation hazard (Weaver et al., 2019), assessing the pre-convective
environment and retrieved atmospheric stability (Iturbide-Sanchez et al. 2018; Bloch et al. 2019; Esmaili et al. 2020),
assessing changes in the intensity of both mid-latitude cyclones and hurricanes (Berndt et al. 2016; Berndt and Folmer
2018) and an evaluation of retrieved soundings for a variety of atmospheric moisture regimes, include regions
impacted by the tropical Saharan Air Layer (Nalli et al. 2016; Kuciauskas et al. 2018).
NUCAPS vertical temperature and moisture soundings were first introduced to National Weather Service forecasters
in 2014. Since then, satellite sounding products have been adapted and expanded in response to end user feedback
(Esmaili et al., 2020). Plan-view and cross-section display capabilities (i.e., Gridded NUCAPS; Berndt et al. 2020) is
one example of a capability developed to facilitate the use of NUCAPS in the operational environment and enable use
for new applications. NUCAPS temperature, moisture, ozone, and derived fields such as TPW are mapped to a 0.5
degree grid with minimal horizontal interpolation utilizing nearest neighbor and vertically interpolated to standard
meteorological levels. Although TPW is not derived by the NUCAPS algorithm, values are calculated with the
standard TPW equation whereby water vapor mixing ratio is vertically integrated from the surface to the top of a
sounding to represent the depth of condensed water vapor in an atmospheric column.
Layer precipitable water (LPW) is the depth of condensed water vapor that exists between two given pressure levels.
An initial LPW product (Forsythe et al. 2015; LeRoy et al. 2016) was developed at CIRA, which employed a fusion
of the NOAA MiRS (Boukabara et al., 2011) water vapor profile retrievals from seven LEO satellites. Satellites used
in the initial LPW study were S-NPP, NOAA-19/20, MetOp-A/B, and Defense Meteorological Satellite Program
F17/18. Advection of retrieved LPW utilizes winds from the Finite Volume Cubed Global Forecast System (FV3GFS,
hereafter GFS) to create Advected Layer Precipitable Water (ALPW), which uses a technique called advective
blending (Gitro et al., 2018). ALPW is computed within four pressure layers and they are (1) surface-850 hPa, (2)
850-700 hPa, (3) 700-500 hPa, and (4) 500-300 hPa. Computation of the ALPW takes place at CIRA and is created
hourly with a 16 km footprint. ALPW allows forecasters to (1) track the movement of water vapor within several
layers and (2) determine the availability of moisture for heavy precipitation events. ALPW complements water vapor
depictions from both geostationary platforms and numerical weather prediction models. Since ALPW is derived from
passive microwave measurements, retrievals are available in the presence of clouds, which is in contrast to infrared-



based retrievals, such as NUCAPS. Water vapor retrievals from MiRS have no dependence on dynamic forecast
models, which allows for independent comparison to model analyses and forecasts.
In order to display both satellite data and retrieved soundings, this study utilized the Advanced Weather Interactive
Processing System (AWIPS). AWIPS was used for the GOES-R satellite product demonstration, which included a
variety of weather applications (Goodman et al., 2012). As such, both ABI and VIIRS satellite imagery, along with
NUCAPS and ALPW products, presented herein, were processed with AWIPS. Further, data transmitted to AWIPS
via the Satellite Broadcast Network has a 6 km footprint on the Full Disk sector; an exception is GeoColor imagery
which is mapped to a 1.5 km grid and available in AWIPS from CIRA via a Local Data Manger feed. Lower
resolution/latency of the full disk sector was suited for imagery of dust within a SAL since the 16-17 February 2020
SAL will be shown to be a slowly evolving feature. Since low latency was not an issue for slowly evolving features,
dust within a SAL was a phenomena that was also well suited to imagery and products from polar orbiting satellites.
**3 Observations from 16-17 February 2020**
During the day of 16 February 2020, a relatively large area of SAL dust moved westward from western Africa to the
eastern Atlantic Ocean. Since there are a dearth of observations over western Africa and the eastern Atlantic Ocean,
this study employed the GFS analyses, or zero hour forecast, to provide supplemental meteorological information.
Superimposed on GOES-16 GeoColor imagery (Miller et al., 2016, 2020) valid at 1800 UTC on 16 February 2020
(Fig. 1), was an inverted trough (Schlueter et al., 2019) at 700 hPa, indicated by a the geopotential heights and a black
dashed contour, and was positioned over the eastern Atlantic. Associated with the inverted trough was a thermal
minimum at 700 hPa, which

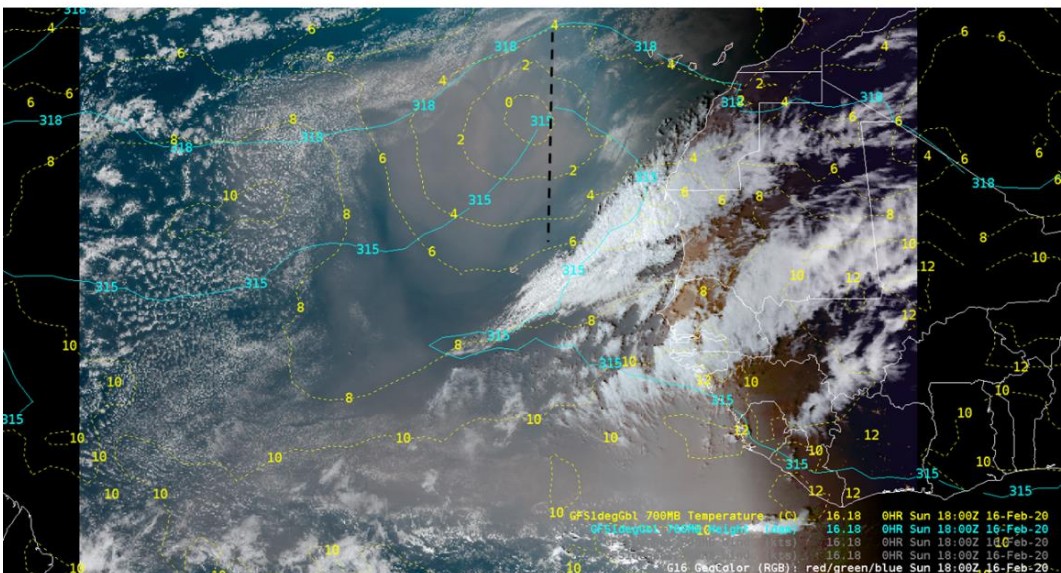

**Figure 1: GeoColor imagery diagnosed from ABI on GOES-16 along with the (1) 315 dm and 318 dm 700 hPa**
**geopotential height (solid contour) and (2) 700 hPa isotherms (˚C, dashed contour) from the GFS analysis. A**
**dashed black contour is used to denote the axis of an inverted trough. All data is valid at 1800 UTC 16 February**
**2020.**
was positioned just west of the trough axis. Temperatures at the center of the thermal minimum were the lowest in the
scene with values near 0˚ C and increased to the southwest to values near 6˚ C. Dust existed in a region that was
roughly bounded by the 315 dm and 318 dm geopotential height contours, to the east by the axis of the inverted trough,
and to the southwest by the 6˚ C isotherm.

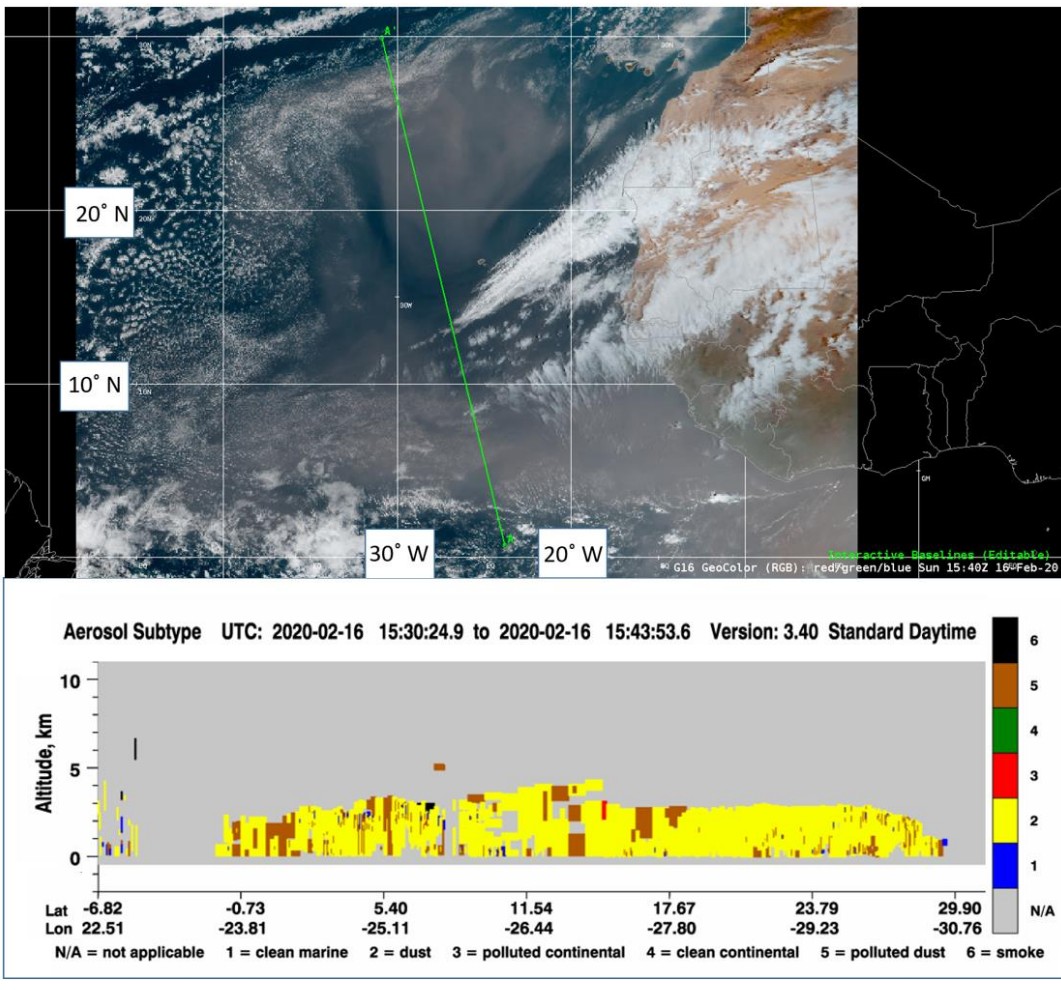

**Figure 2: GeoColor imagery diagnosed from ABI on GOES-16 valid at 1540 UTC 16 February 2020 along with**
**a portion of the ground track (green line segment) of CALIPSO from 1530 UTC to 1543 UTC 16 February**
**2020. Retrieved aerosol subtype is displayed in the vertical feature mask from CALIOP.**



There was an ascending CALIPSO over-passes located over the eastern Atlantic Ocean for the dust case discussed
herein. Data from CALIOP provided information about the field of aerosols evident in Fig. 1. In Fig. 2a, the orbit of
CALIPSO, denoted by a green line segment, entered the scene along the southern portion of the image at
approximately 1530 UTC 16 February 2020, moved towards the northwest and exited the scene at approximately 1543
UTC 16 February 2020. As stated in Section 2, the VFM provides information about atmospheric constituents in a
vertical plane. One of the constituents was aerosol, which was further sub-classified into four aerosol types: dust,
polluted continental, polluted dust, and smoke. As can be seen in Fig. 2b, there were two sub-aerosol types in the
lower atmosphere from the surface to approximately 3.0 km over a significant portion of the orbit in Fig. 2a. North of
approximately 15˚ N, VFM suggested dust was the primary constituent in the aerosol layer. However, the VFM
suggested a significant portion of the aerosol layer, south of 15˚ N, was occupied not only by dust, but also by polluted
dust.
Although data contained in the VFM image of Fig. 2b is a vertical cross section along the orbital path of CALIPSO,
a few assumptions are made herein. This manuscript will hypothesis that all aerosol in the region north of
approximately 15˚ N and west of the inverted trough in Fig. 1 was dust and will be referred to as the northern region
of dust; henceforth NRD. Further, this manuscript will also speculate that all aerosol in the region south of about 10˚
N and west of Africa was a mixture of pollution dust and dust and will be referred to as the southern region of dust;
henceforth SRD.

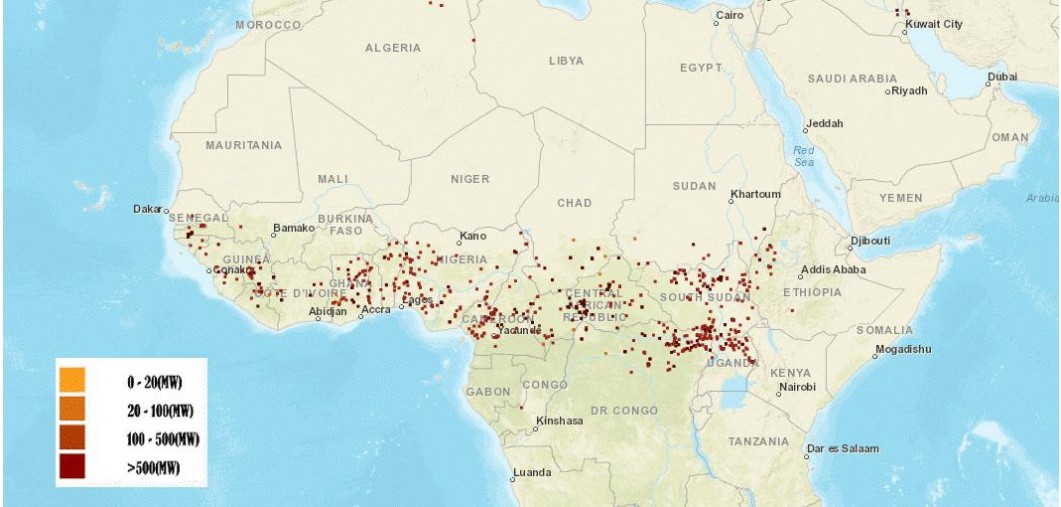

**Figure 3: VIIRS active fire map for 15 February 2020, one day prior to the dust case study herein. Dots indicate**
**the locations of burning while the color of the dots denotes values of the fire radiative power. Credit is given to**
**the VAFM group for the image.**
Two questions arise about the aerosol layer in Fig. 2. First, can an inference be made about the vertical depth of the
entire dust field based on the VFM? Perhaps, results from Adams et al. (2012) provides a climatology of the vertical
depth of SAL dust, which suggests in the months December, January, and February, dust layers from western Africa
are about 2.0 to 3.0 km thick (see their Fig. 3c). Second, examination of Fig. 2a exhibits an east-west layer of aerosol
south of about 10˚ N. Is there a source of pollution that can help explain the existence of polluted dust south of 10˚ N
in Figs. 2a and 2b? One possible candidate is smoke from biomass burning along equatorial Africa.
One of the products from the VIIRS instrument is the VIIRS Active Fire Map (VAFM) (Csiszar et al., 2014). A plot
of the VAFM from 15 February 2020, one day prior to the dust case discussed herein, is displayed in Fig. 3. Regions
of active fire were indicated by dots of varying colors; each color represents a range of values of the fire radiative
power, which can be used to estimate emissions from biomass burning (Ahmadov et al., 2017). Biomass burning
occurred in the latitudinal range from the Equator to about 10˚ N over Africa. This manuscript speculates that smoke
from burning on 15 February 2020, indicated in Fig. 3, may be the source of pollution of the polluted dust retrieved
by CALIOP south of 10˚ N in Fig. 2b.
There were two main regions of aerosol in satellite imagery on 16 February 2020.  To begin with, a black oval is used
in Fig. 4 to demark the NRD seen in GeoColor imagery from ABI onboard GOES-16; similarly, the SRD is denoted
by a broken, east-west, black line segment in Fig. 4. There are also a few additional annotation symbols in Fig. 4 that
will be discussed shortly.

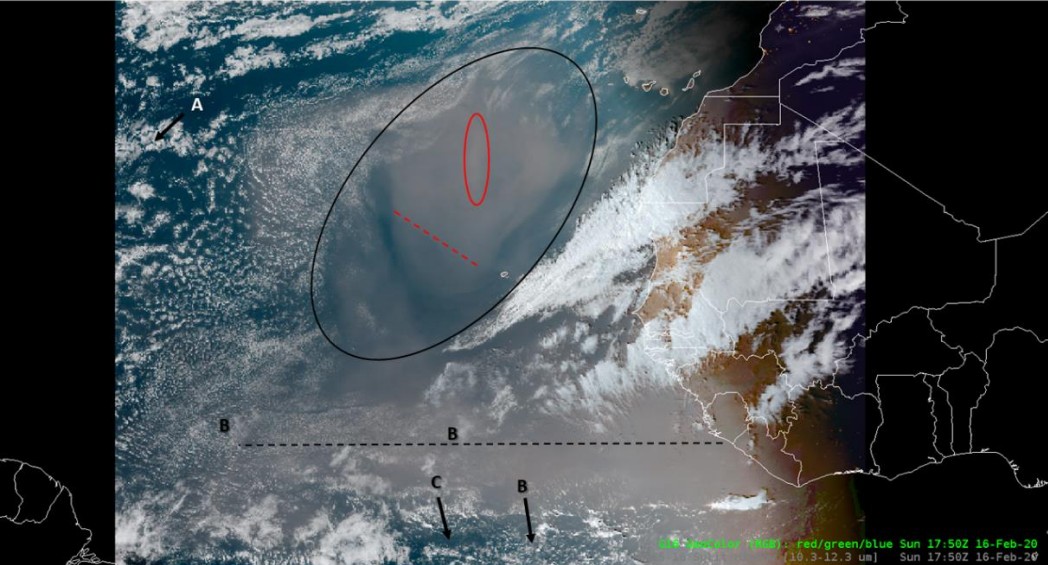

**Figure 4: GeoColor imagery diagnosed from ABI on GOES-16, valid 1800 UTC 16 February 2020, along with**
**the following annotations: A black oval bounds dust in the northern dust region while the horizontal, black,**
**dashed line highlights dust in the southern dust region. Within the black oval are additional annotations in red.**
**Further the letters A (upper left portion of the figure), B, and C appear. All annotations are used for**
**comparison purposes with Fig. 5.**
As mentioned in the introduction, channel differencing of infrared channels has been used, at times, to identify lofted
dust. In a study by Miller et al. (2019), several numerical experiments were used to examine channel differencing of
infrared wavelengths and dust detection. Specifically, plots of values of brightness temperatures (Tbs) at 12.3 µm
subtracted from values of Tbs at 10.35 µm, Tb(10.35 µm) – Tb(12.3 µm), were shown to be negative for airborne
dust, with a caveat: Vertically integrated values of water vapor had to be below some critical value. In contrast, if
values of water vapor in a layer, which also contained dust, exceed a critical integrated amount, then values of the
channel difference were shown to be positive. Although GeoColor imagery is a novel way to display imagery (Fig.
4), horizontal variations of vertically integrated water vapor has little impact on such imagery. Horizontal variations
of vertically integrated water vapor do impact channel differencing between Tbs at 12.3 µm and 10.35 µm, which is
displayed in Fig. 5, valid at 1800 UTC 16 February 2020.

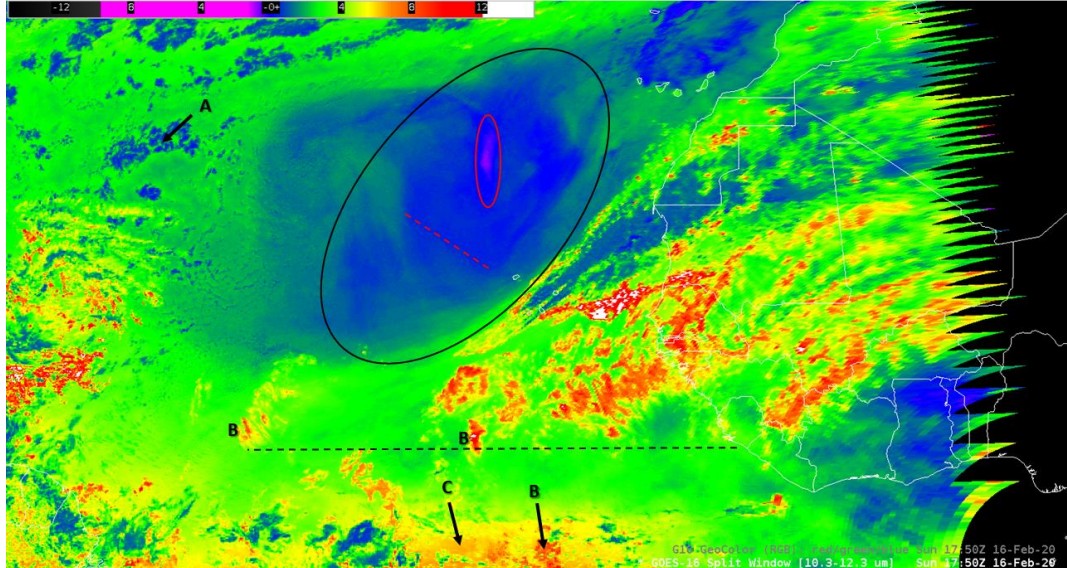

**Figure 5: Channel difference, Tb(10.35 µm) - Tb(12.3 µm) (°C), from ABI on GOES-16 valid at 1800 UTC 16**
**February 2020. Annotations are the same as in Fig. 4. Dust is indicated by the blue and purple colors within**
**the black oval in the northern dust region. There was a lack of a dust signal in the southern dust region.**
Most of the region bounded by the black oval in Fig. 5 had values of the channel difference near zero (blue) and less
than zero (purple). A word of caution is warranted: There does not exist, in general, a functional mapping between an
atmospheric feature and a value of the ABI channel difference Tb(10.35 µm) – Tb(12.3 µm). In other words, there
does not exists an inverse mapping from a value in the channel difference to a unique atmospheric feature. With that,
two features may be mapped to the same value of the channel difference. For example, low-level liquid water clouds





and dust were both mapped to values that were near zero or negative while thin cirrus and relatively moist boundary
layers were both mapped to relatively large positive values (orange/red).
Physical interpretation of values in the channel difference image was be done by direct comparison with a GeoColor
image. Two features appeared blue in Fig. 5; one feature was within the black oval while another was located in the
upper left portion of the figure, which is denoted by a white colored letter "A". A direct comparison of these features
between the GeoColor image in Fig. 4 and the channel difference in Fig. 5 suggested that the blue color within the
oval (Fig. 5) was associated with the dust plume (Fig. 4) while the blue color in the upper left of Fig. 5 was associated
with low-level liquid water clouds in Fig. 4. Note also the difference in the appearance of the edge of the blue regions
in Fig. 5. That is, dust had a boundary that appeared diffuse; however, liquid water clouds had a rather sharp contrast
with the environment at their boundary in Fig. 5. There were also two features that appeared orange/red in Fig. 5. One
feature was located slightly above the middle and left edge of the black, broken, horizontal line segment, denoted by
the letter "B" along with a region along the lower edge of the figure; also denoted with a letter "B" and arrow. Both
orange/red features were barely discernable in Fig. 4 as they were thin cirrus. A second feature was located along the
bottom of Fig. 5, denoted by the letter "C", and appeared as a somewhat homogeneous orange color, which was
coincident with clear skies in Fig. 4. Physical interpretation served to illustrate the lack of a functional mapping
between an atmospheric feature and a value of the channel difference. This manuscript will now focus on the dust
plume as seen in black oval exhibited in Figs. 4 and 5.

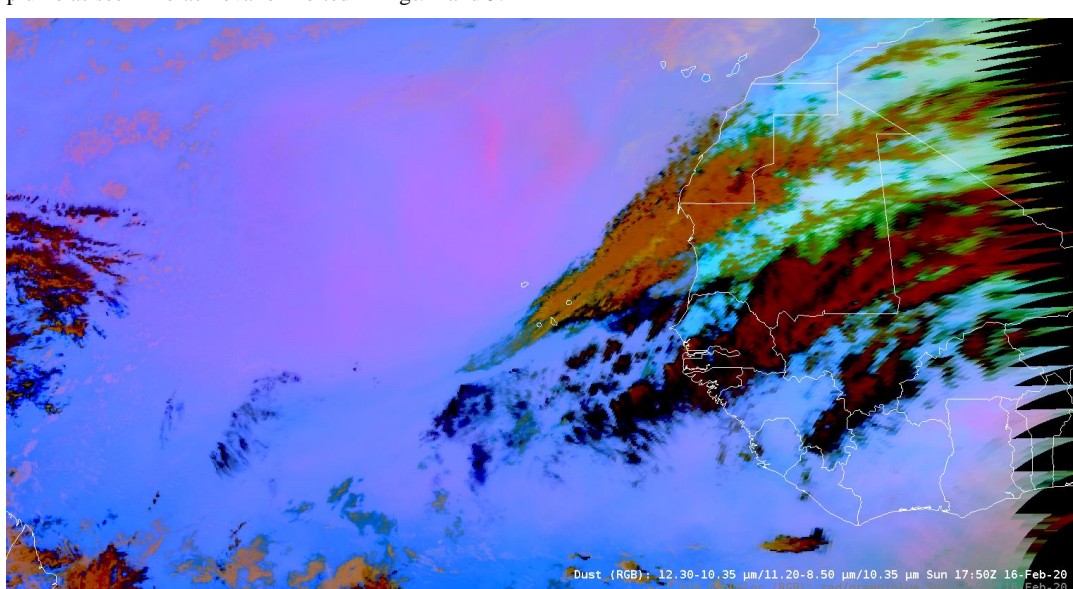

**Figure 6: SEVIRI dust product generated by using ABI bands on GOES-16 on 1800 UTC 16 February 2020. A**
**pink color, indicating dust, was characteristic of the northern dust region while a blue color was characteristic**
**of the southern dust region; that is, dust within the southern dust region was less obvious.**



Additional annotations appear within the black oval of Figs. 4 and 5. To begin with, the red oval, in Fig. 4, bounded
a north-south region of dust. Just to the northwest and southeast of the red oval, GeoColor imagery exhibited a touch
of blue, suggesting that the dust was unable to obscure the ocean when compared to the region of dust within the red
oval. A similar pattern was evident in the colors of the channel difference (Fig. 5). That is, the region within the oval
contained negative values (blue/purple) of the channel difference while just to the northwest and southeast of the red
oval positive values (green) appeared. Similarly, there was an additional subtle touch of blue, within the dust field, to
the southwest of the broken red line segment in Fig. 4 compared to regions to the northeast of the broken red line
segment within the black oval. Similarly, increased values of the channel difference (blue/green) existed to the
southwest of the broken line segment while smaller values (blue) existed to the northeast of the segment in Fig. 5.
That the horizontal variability of appearance of dust in the black oval of Fig. 4 corresponded to a similar horizontal
variability of values of the channel difference in the black oval of Fig. 5 supported the assumption that the blue/purple
regions within the black oval in Fig. 5 was in response to the dust seen in Fig. 4. Note also a lack of a dust signal along
the broken, east-west, black line segment in Fig. 5. That is, values of the channel difference were positive with values
near 3° C to 4° C. Although the Tb(10.35 μm) - Tb(12.3 μm) channel difference is a component of a dust product,
there are other components.
A dust product, which was developed for the SEVIRI instrument onboard MSG (Ashpole and Washington, 2012),
was adapted to ABI bands for the dust case herein. Tbs from three of the ABI bands were used to generate the SEVIRI
dust product and they are 8.5 μm, 10.35 μm, and 12.3 μm. A multicolor image was generated by assigning values of
Tb(12.3 μm) - Tb(10.35 μm) to red, Tb(11.20 μm) - Tb(8.5 μm) to green, and Tb(10.35 μm) to blue (Fig. 6). As a
result of the SEVIRI dust recipe, dust is indicated by a pinkish color. Due to differences in the spectral width and
central wavelength of ABI bands compared to SEVIRI bands, the color component thresholds were adapted to account
for the differing spectral characteristics and maintain the appearance of the dust product generated with ABI bands
compared to SEVIRI bands (Shimizu 2015, Berndt et al. 2018). For example, the ABI band centered near 10.35 μm
has a spectral width from 10.1 μm to 10.6 μm—0.5 μm; in contrast, the SEVIRI band centered near 10.80 μm has a
spectral width from 9.8 μm to 11.8 μm—2.0 μm. Further, the spectral width for the SEVIRI band centered near 10.80
μm exhibits an overlap with the SEVIRI band centered near 12.0 μm, which has a spectral width from 11.0 μm to
13.0 μm.

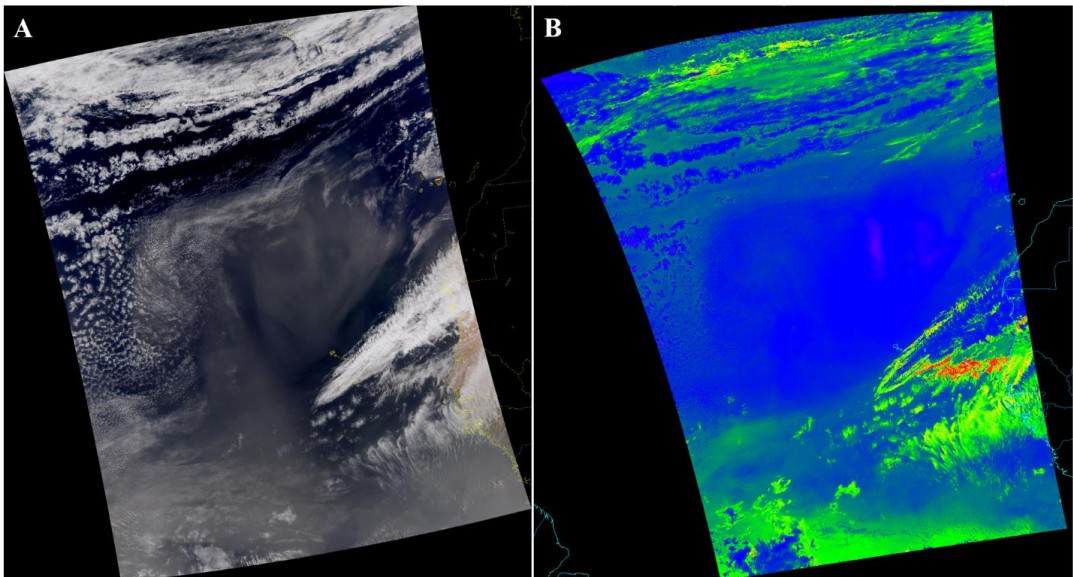

**Figure 7: Data from VIIRS on NOAA-20 valid at approximately 1510 UTC 16 February 2020 showing (a) True-Color, as opposed to GOES-16 ABI GeoColor, imagery and (b) VIIRS channel difference, Tb(10.76 μm) - Tb(12.01 μm), with the same color table shown in Fig. 5.**

As stated earlier, the channel difference Tb(10.35 μm) - Tb(12.3 μm) is used in dust detection algorithms. Consequently, horizontal variations evident in the NDR in Fig. 5 were also evident in Fig. 6. One of the assumptions stated above was that all aerosol north of approximately 15˚ N and west of the inverted trough, the NDR in Fig. 4, was dust. An examination of the NDR in Fig. 6 exhibited bright pink colors, which provided support for the first hypothesis. A second assumption stated that all aerosol south of about 10˚ N and west of Africa was a mixture of smoke and dust, the SDR in Fig. 4. In contrast to the NDR, an examination of the SDR in Fig. 6 exhibited mostly blue colors with a few hints of pink in some regions. Even though CALIOP data suggested dust in the SDR, support for the second hypothesis, based on results in Fig. 6, is less obvious. A question arises, why was there higher confidence of dust in the NDR compared to the SDR? Banks et al. (2019) and Miller et al. (2019) explored a similar question; results from their numerical studies suggested that water vapor, in excess of a critical value, may mask dust.

There was also a NOAA-20 overpass in the region of interest at approximately 1510 UTC 16 February 2020, just prior to the CALIPSO overpass. As a reminder, GeoColor imagery from ABI is produced from the following three bands: 0.47 μm, 0.64 μm, and 0.87 μm. These three bands are then used to diagnose values of green reflectances; GeoColor imagery is produced by combining the ABI red, diagnosed green, and ABI blue bands. Since VIIRS measures radiances within the red, green, and blue regions of the electromagnetic spectrum, a true-color, as opposed to GeoColor, image was produced for the dust case herein (Fig. 7a). Although the VIIRS true-color image was captured near 1510 UTC, there existed a similarity of aerosol features to the 1800 UTC ABI GeoColor image in Figs. 1, 2, and



4. Although there was a nearly three hour difference between the VIIRS and ABI images, similarity of aerosol features
suggested a relatively slow temporal morphology of dust in the NDR and SDR.
VIIRS also contains bands from which the channel difference, a companion to Fig. 5, was generated. In Fig. 7b, the
channel difference is shown with the same color table in Fig. 5. A comparison between Figs. 5 and 7b reveals that,
despite the same color table, the two channel difference images were different. While the channel difference in Fig. 5
was made with Tb(10.35 µm) - Tb(12.3 µm), the channel difference in Fig. 7b was made with Tb(10.76 µm) - Tb(12.01
µm). That is, the central wavelengths used in the channel difference for ABI and VIIRS were different. Further, the
spectral widths of each band were also different. For example, the ABI spectral width for Tbs near 10.35 µm ranged
from 10.1 µm to 10.6 µm; the VIIRS spectral width for Tbs near 10.76 µm ranged from 10.26 µm to 11.26 µm.
Qualitatively, interpretation of the horizontal pattern of values of the channel difference in Fig. 7b were similar to the
interpretation of Fig. 5. That is, the NDR was characterized by values of the channel difference that were less than
zero while the SDR was characterized by larger values of the channel difference. In other words, there was more of a
dust signal in the NDR compared to the SDR.
Efforts will now focus on vertically integrated tropospheric water vapor for the scene in Fig. 1. There exist
measurements from ABI on GOES-16 which are in response to vertically integrated water vapor: Band 10, which
detects upwelling radiation centered near 7.34 µm and is referred to as the low-level water vapor band. Low-level
water vapor imagery at 1800 UTC 16 February 2020 is displayed in Fig. 8. In the low-level water vapor image,
maximum values of Tbs were in the yellow region with values near -2° C. Values of Tbs decreased to near -15° C in
the region of orange-red; then continued to decrease to values near -18° C in regions of orange. In particular, values
of Tbs decreased from approximately -2° C (yellow) to -18° C (orange) both north and south of the Tb maximum;
stated differently, Tbs decreased about 15° C from the Tb maximum to the north-northeast and southeast. Was the
decrease in values of Tbs of the low-level water vapor image due to horizontal variations of air temperature?

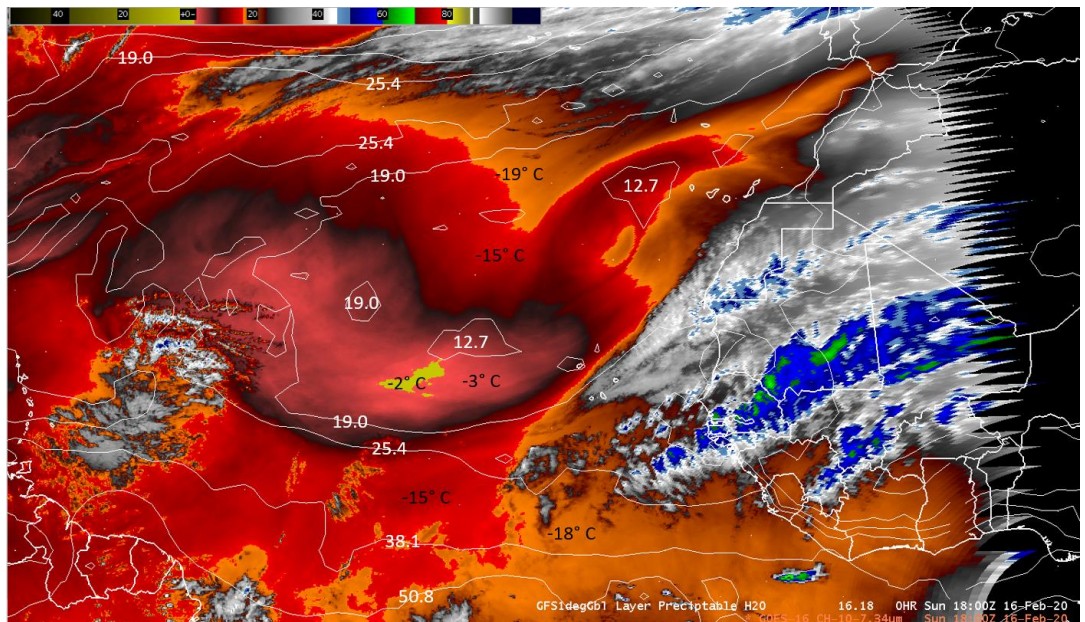

**Figure 8: Tb(7.34 µm) from ABI on GOES-16, color shaded with a few numerical values, along with TPW (mm;**
**white contours) from the GFS analysis and a few values of Tb(7.34 µm) in black. All data is valid at 1800 UTC**
**16 February 2020.**
Brightness temperatures in the low-level water vapor image are presently compared to air temperatures. Since the
weighting function for 7.34 µm imagery generally peaks at pressures greater than approximately 500 hPa (Schmit et
al., 2018), air temperatures from 700 hPa to 500 hPa were examined. Isotherms at 700 hPa and 500 hPa, from the GFS
00 hour forecast, were plotted on ABI 7.34 µm imagery; all data was valid at 1800 UTC 16 February 2020 (Fig. 9).
As seen in Fig. 9, values of the air temperature at 700/500 hPa were approximately 8˚ C/-7˚ C near the maximum
value of Tbs near 7.34 µm (yellow). Values of the air temperature then decreased to near 6˚ C/-11˚ C, to the north-
northeast of the Tb max where values of the Tb were near -18˚ C. Likewise, values of the air temperature at 700/500
hPa increased from about 8˚ C/-7˚ C to near 10˚ C/-4˚ C to the southeast of the Tb maximum. Note the lateral change
in values of Tbs were approximately 15˚ C; in contrast, the lateral change in values of the 700/500 hPa air temperature
were, in absolute value, about 2˚ C/4˚ C. As a reminder, one characteristic of the tropical atmosphere is that
geopotential variations and horizontal temperatures gradients are relatively small (Holton, 1979). Consequently,
lateral changes in air temperature were unable to explain the lateral changes in Tbs: Another reason was sought.
Values of TPW from the GFS are shown in relation to various satellite fields. In Fig. 8, values of TPW were plotted
on the low-level water vapor image; both valid at 1800 UTC 16 February 2020. In general, regions with the smallest

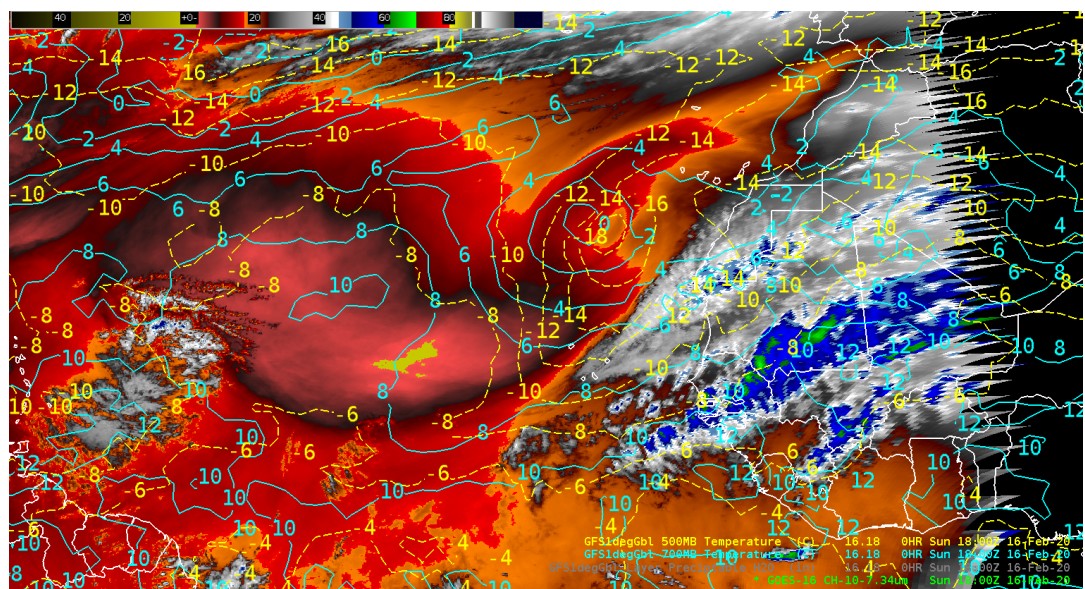

**Figure 9: Same as Fig. 8, except contoured values of 700 hPa (solid) and 500 hPa (dashed) temperatures (°C) from the GFS analysis are plotted; all data is valid at 1800 UTC 16 February 2020.**

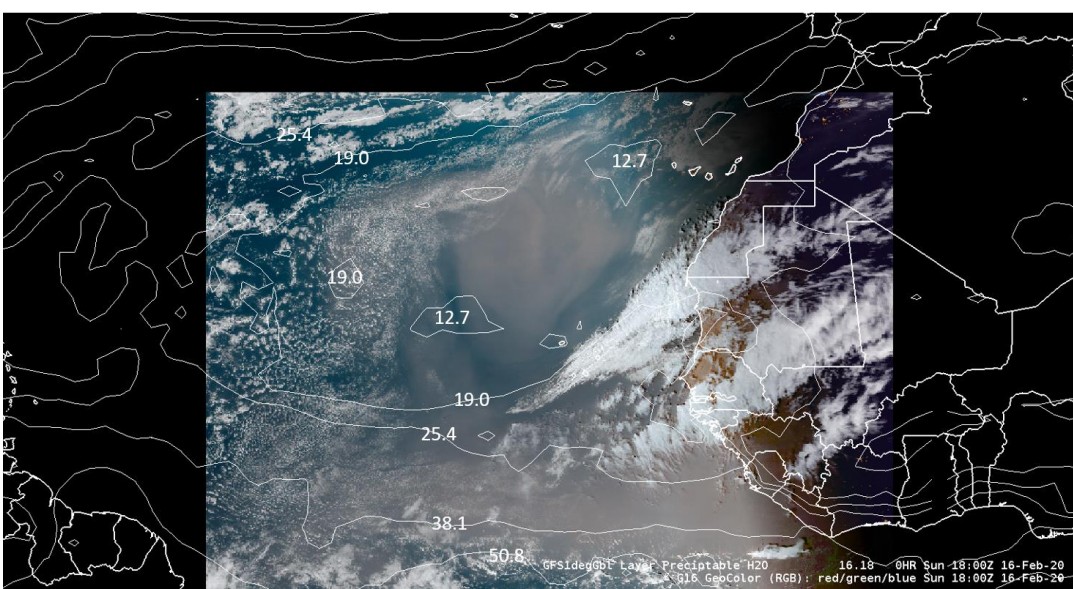

**Figure 10: TPW (mm) from the GFS analysis plotted on a GeoColor image derived from GOES-16 ABI; all data is valid at 1800 UTC 16 February 2020.**

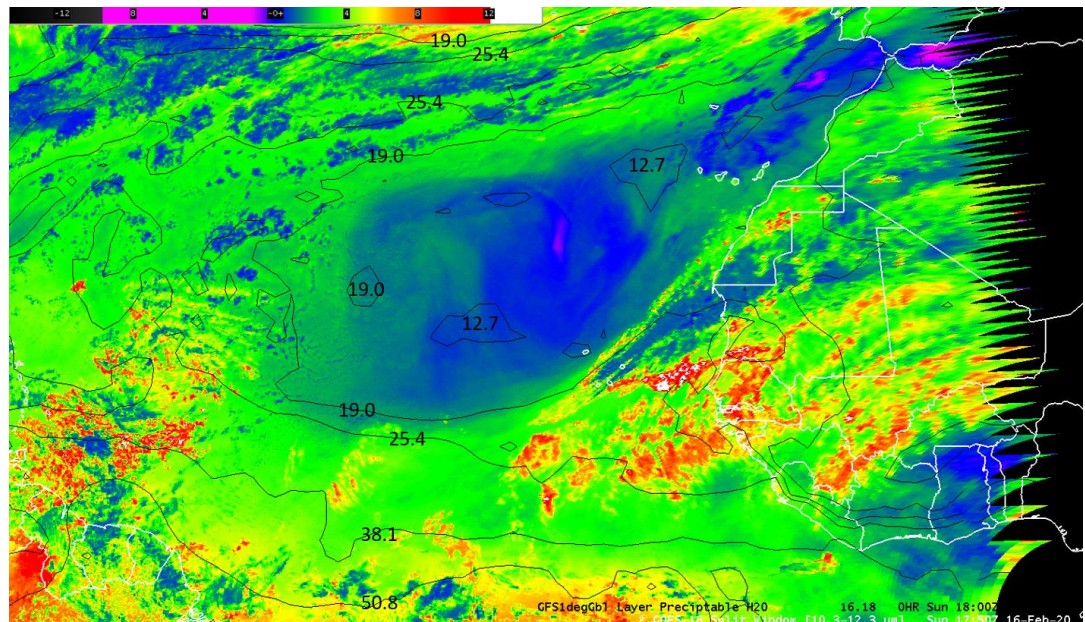

**Figure 11: Same as Fig. 10, except TPW (mm) is plotted on the GOES-16 ABI Tb(10.35 μm) - Tb(12.3 μm)**
**channel difference (˚C).**

values of TPW corresponded to regions with the largest values of Tbs in the low-level water vapor image. However,

notice that the region to the north northeast of the Tb maximum, values of Tbs were approximately -19˚ C in a location

with values of the TPW near 19 mm. In contrast, values of Tbs to the southeast of the Tb maximum were also near -

19˚ C; however, values of TPW were approximately double with values about 38 mm. One possible reason for greater

values of TPW to the southeast of the Tb maximum, compared to the north northeast of the Tb maximum, was that

more water vapor existed below the peak of the weighting function for 7.34 μm. That is, values of boundary layer

water vapor decreased from regions to the southeast of the Tb maximum to regions to the north northeast of the Tb

maximum. Values of analyzed surface dewpoint temperatures from GFS at 1800 UTC 16 February 2020 suggested

that values of dewpoint temperatures decreased from near 21˚ C, southeast of the maximum value of Tb near 7.34 μm,

to values near 15˚ C, to the north northeast of the maximum value of Tb near 7.34 μm. Values of TPW are also

displayed on both a GeoColor image (Fig. 10) and the channel difference (Fig. 11). Not only was the region of the

NDR in the GeoColor image co-located with the smallest values of TPW (Fig. 10), but also the region of values of the

channel difference that were near and less than zero, and associated with the NDR, were also co-located with the

smallest values of TPW (Fig. 11). In contrast, the SDR was associated with values of the TPW that were approximately

two to three times larger than values of the TPW associated with the NDR. There existed additional satellite sensors

that had the ability to address water vapor in the atmosphere.

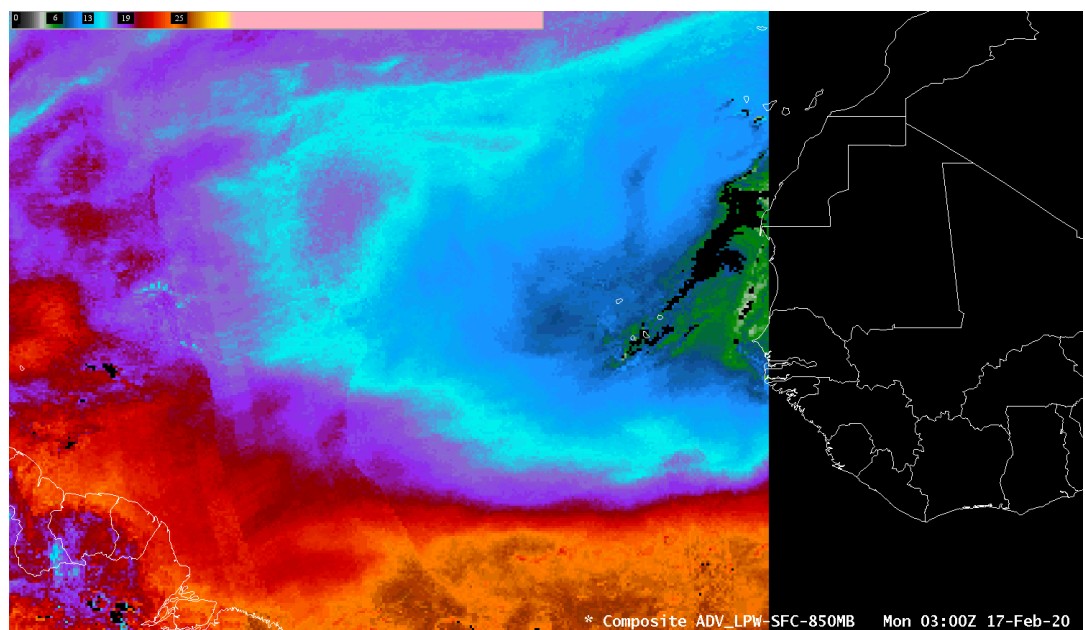

Figure 12: Advected Layer Precipitable Water product (mm) for the surface to 850 hPa layer valid at 0300 UTC 17 February 2020.

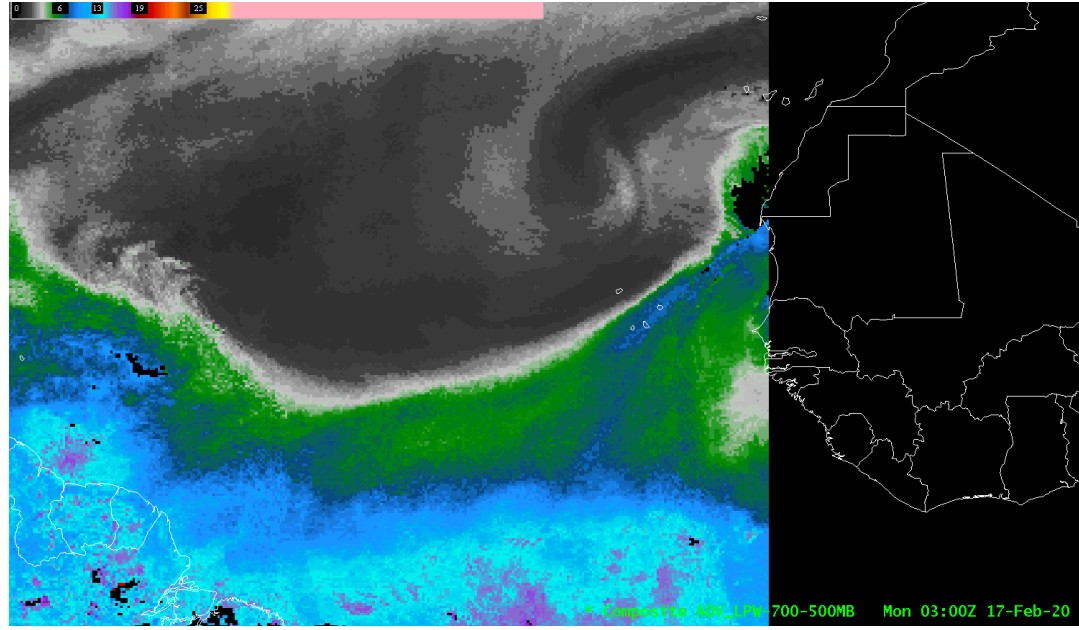

Figure 13: Same as Fig. 12, except for the 700 hPa to 500 hPa layer valid at 0300 UTC 17 February 2020.





One of the uses of microwave data is the retrieval of water vapor. As stated above, Forsythe et al. (2015) developed a
methodology to retrieve values of water vapor in layers of the troposphere, referred to as the ALPW product. Due to
the use of LEO sensors, imagery for the ALPW was not necessarily available as often as ABI data from GOES-16.
Subsequently, retrieved values of ALPW, in the layer from the surface to 850 hPa, valid at 0300 UTC 17 February
2020 are displayed in Fig. 12. Previously, this manuscript speculated that values of dewpoint temperature at the surface
decreased from south to north, relative to the Tb maximum in Fig. 8. Values of the ALPW, in Fig. 12, support the
speculation; that is, values of the ALPW decreased from the south, with values approximately 27.9 mm, to the north,
with values near 15.4 mm.

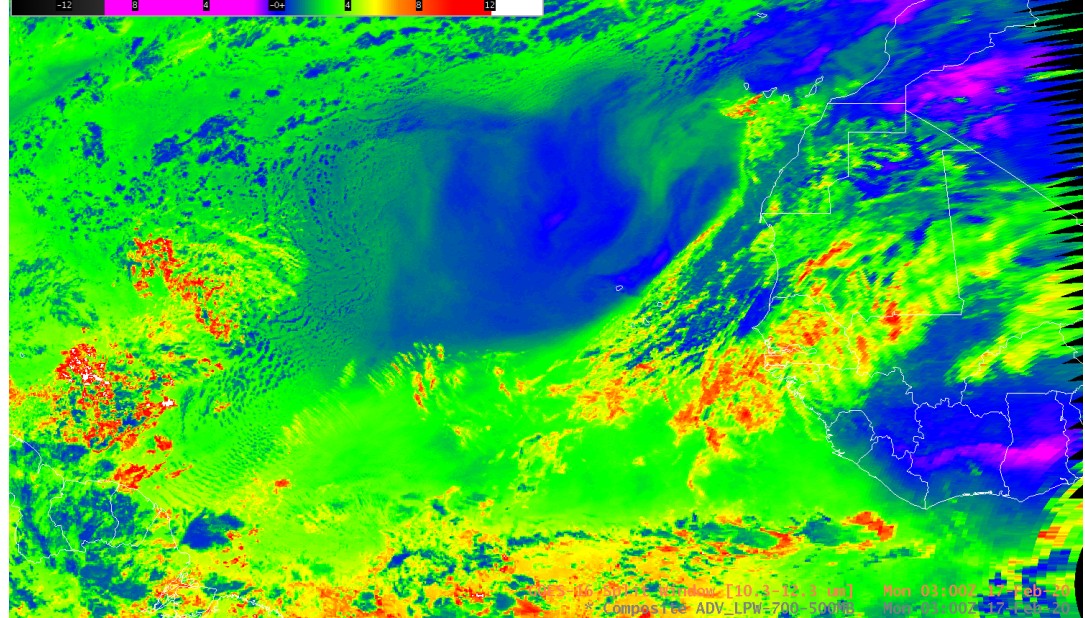

**Figure 14: GOES-16 ABI Tb(10.35 μm) - Tb(12.3 μm) channel difference (˚C) valid at 0300 UTC 17 February**
**2020.**
A second layer of values of the ALPW from 700 hPa to 500 hPa is shown in Fig. 13. Two characteristics of values of
the ALPW in the second layer were (1) a relatively large region of the northern half of the image had values near 2.5
mm and (2) values of the ALPW increased towards the south to values near 12.7 mm. Further, the boundary between
dark grey and green was co-located with the southern edge of the region of largest values of Tb near 7.34 μm where
values of GFS TPW increase from about 19 mm to near 25 mm (Fig. 8). Note also that the region of the NDR in the
GeoColor image (Fig. 10) was co-located with the smallest values of ALPW in Figs. 12 and 13. In contrast, values of
the ALPW in Figs. 12 and 13, increased in the SDR, particularly in Fig. 13. In addition, values of the channel difference
displayed in the same scene as Figs. 12 and 13 also showed a dust signal (Fig. 14) that was also co-located with the
smallest values of ALPW in the NDR in Figs. 12 and 13. Contrariwise, values of the channel difference increased to
values near 3˚ C to 4˚ C in the SDR, similar to values of the channel difference, nine hours earlier, in Fig. 5.
In addition to the retrieval of values of ALPW, data from NUCAPS was used to diagnose TPW. Since data from LEO
satellites are used by the NUCAPS algorithm, gridded values of TPW are shown in Fig. 15, valid at 0333 UTC 17
February 2020, which was the time of the granule shown in Fig. 15. A large region of values of TPW north of
approximately 15˚ N, in the NDR, varied between 12 mm and 16 mm.  In sharp contrast, values of TPW south of
approximately 15˚ N, in the SDR, increase over a relatively short distance to values in excess of 26 mm. In order to
further examine the three dimensional structure of the scene shown in Fig. 15, retrieved NUCAPS soundings were
examined.

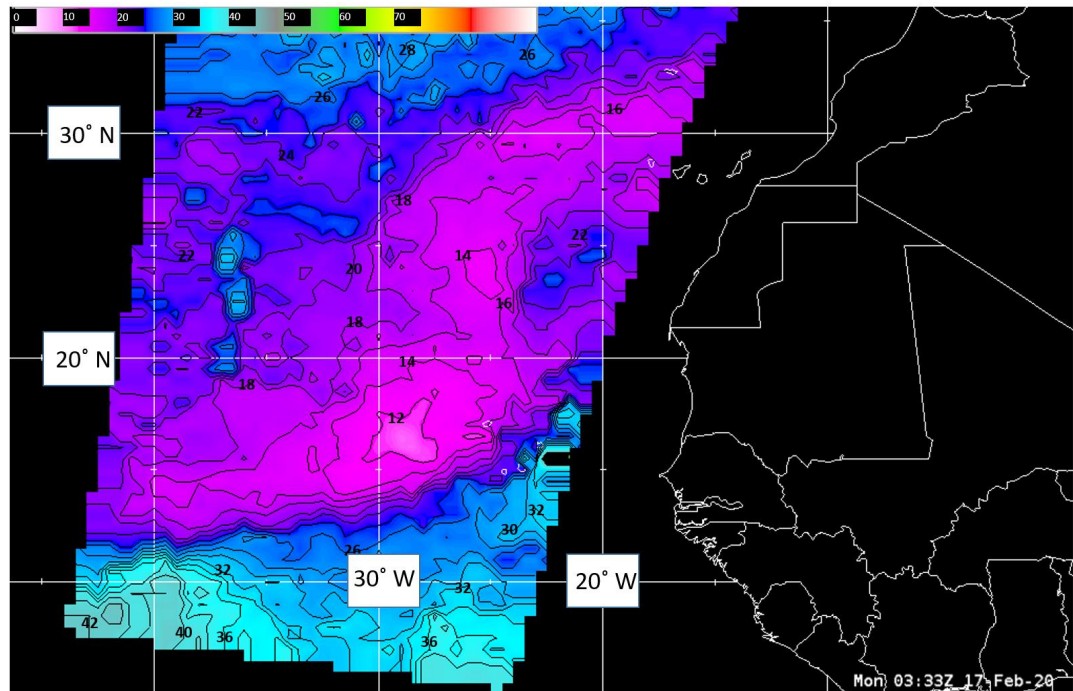

**Figure 15: Gridded NUCAPS TPW (mm) valid at 0333 UTC 17 February 2020, which represents the time of**

**the granule in the image.**

A swath of locations of NUCAPS soundings is shown in Fig. 16. NUCAPS sounding locations, indicated by green
dots, were superimposed on an ABI low-level water vapor image in order to relate soundings to the NDR and SDR.
Both datasets were valid at 0240 UTC 17 February 2020, which was the beginning time of the orbit. Nine soundings
from NUCAPS in the NDR were examined, the location of a representative sounding site is bounded by a white circle
and identified by the numeral 1 and will be referred to as Sounding-1. Similarly, three soundings from NUCAPS were
examined in the SDR, a circle bounds the location of a representative sounding site and is denoted by the numeral 2,
and will be referred to as Sounding-2. Since the horizontal areal extent of the NDR, in Fig. 16, was larger than the
horizontal extent of the SDR, more NUCAPS soundings were used to sample the NDR compared to the sample size





of the SDR. Both NUCAPS retrieved soundings, Sounding-1 and Sounding-2, are displayed in Fig. 17. A noticeable
characteristic of Sounding-1 was the relatively large value of the difference between the temperature and the dewpoint
temperature of several tens of degrees Celsius, especially in the layer between 700 hPa and 500 hPa, which was
coincident with the weighting function peak of the low-level water vapor image (Fig. 8). All of the other eight
NUCAPS soundings in the NDR contained a similar difference between the temperature and dewpoint temperature.
In contrast, however, values of the difference between the temperature and dewpoint temperature in Sounding-2 were
on the order of ten degrees Celsius; a characteristic shared by the other NUCAPS soundings in the SDR. Observations
suggested that the relatively low water vapor content characteristic of the NDR compared to the SDR allowed dust to
be detected by the ABI Tb(10.35 µm) - Tb(12.3 µm) difference in the NDR compared to the SDR.

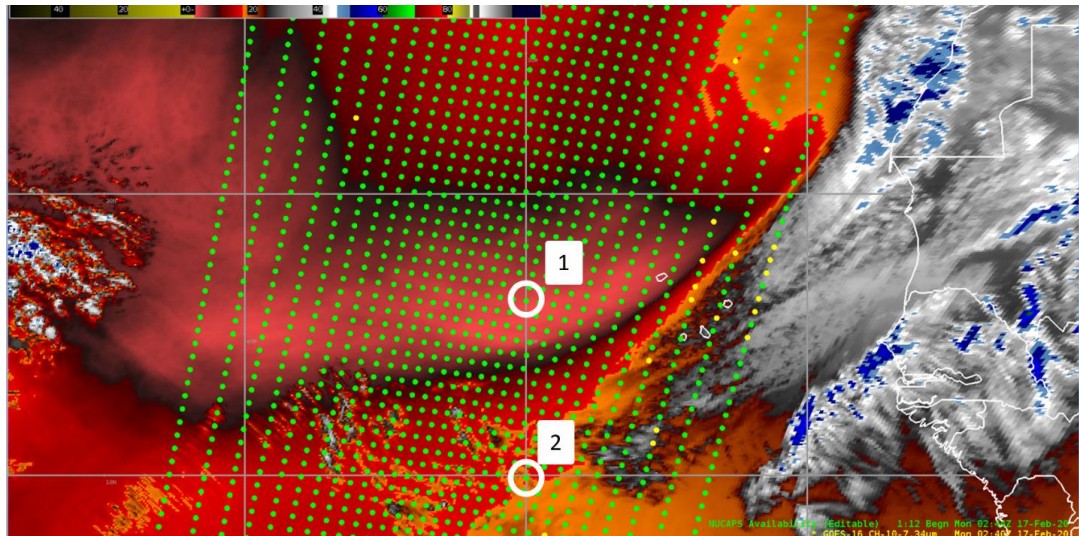

**Figure 16: Low-level water vapor image from GOES-16 ABI along with a swath of locations (dots) of NUCAPS**
**soundings, both valid at 0240 UTC 17 February 2020. Numerals 1 and 2 denote locations from which retrieved**
**NUCAPS soundings were extracted.**
There existed important consequences of relatively low water vapor content characteristic of the NDR compared to
the SDR. First, the ABI Tb(10.35 µm) - Tb(12.3 µm) difference had values that were negative to near zero, which was
consistent with dust in the NDR. That is, relatively low values of water vapor allowed dust to be detected. One counter
argument against the use of the ABI infrared channel difference is that dust was evident in GeoColor images, which
appeared earlier in the manuscript. A close examination of Fig. 10, however, shows that the day/night terminator was
near the western coast of Africa at 1800 UCT 16 February 2020. Once solar reflection ceased over the eastern Atlantic
Ocean, GeoColor imagery was unable to reveal future locations of dust in the NDR. Thus, a second consequence of
relatively low water vapor content was that the dust field in the NDR may be tracked in time, without solar reflection,
which was demonstrated in a time sequence of night-time images of values of the ABI Tb(10.35 µm) - Tb(12.3 µm)
illustrated in Fig. 18. One feature was highlighted from 2000 UTC 16 February 2020 to 0200 UTC 17 February 2020





in Fig. 18: Over the six hour time period, the horizontal pattern of the channel difference exhibited little change, which
supports the relatively slow temporal morphology of the dust contained in the discussion for Fig. 7a.

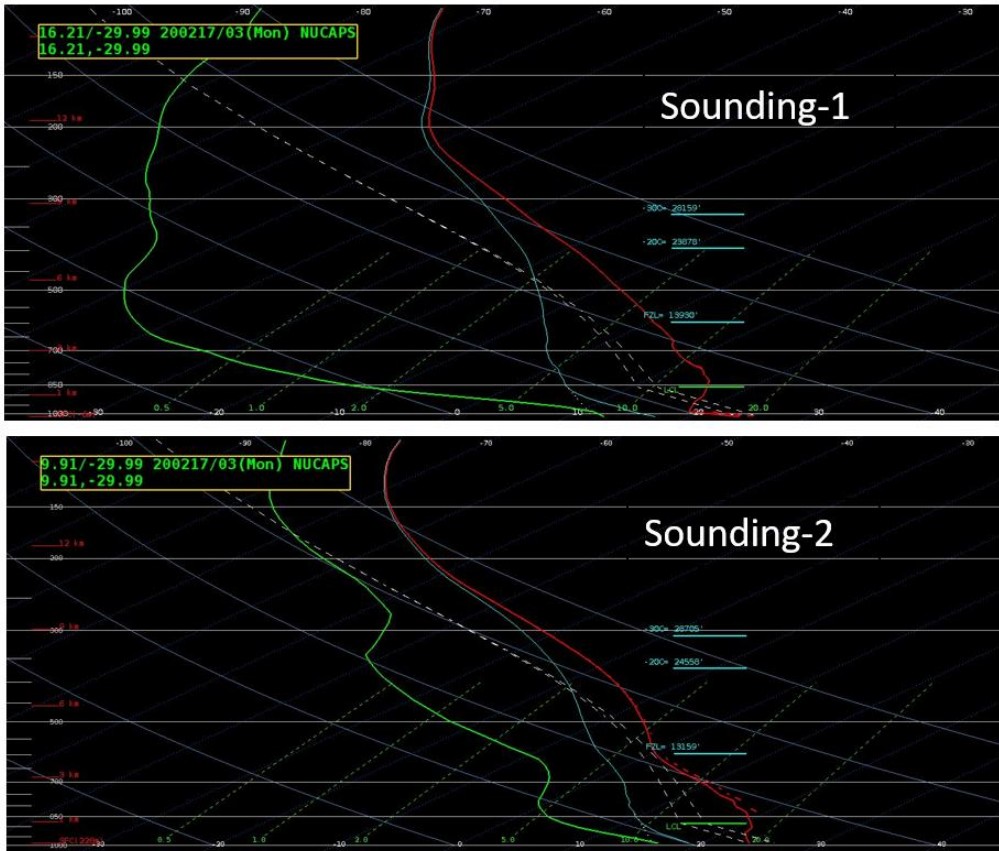

**Figure 17: Atmospheric soundings, denoted Sounding-1 and Sounding-2, from the location identified by the**
**numerals 1 and 2 of Fig. 16. Sounding-1 illustrates the dry atmosphere characteristic of the northern dust**
**region compared to an atmosphere that was moister and characteristic of the southern dust region.**
**4 Role of Data Assimilation**
There has been increasing research efforts to assimilate dust, or aerosol in general, into numerical models for the
improvement of aerosol weather forecast over the last two decades (Collins et al., 2001, Wang et al., 2003, Weaver et
al., 2007, Wang and Niu, 2013, Zhang et al., 2014, Lee et al., 2017). In addition to research efforts, many operational
numerical weather prediction (NWP) centers have included aerosols in their assimilation systems to provided routine
aerosol forecast and aerosol re-analysis (Xian et al., 2019). A brief list of some NWP centers and their efforts are the
following: the US Navy Fleet Numerical Meteorology and Oceanography Center (FNMOC), which employs the Navy
Aerosol Analysis and Prediction System (NAAPS) to provide reanalysis (Lynch et al., 2016) and ensemble forecast



(Rubin et al., 2017) of aerosol distribution; the European Centres for Medium-Range Weather Forecasts (ECMWF),
which utilizes a 4DVar data assimilation algorithm to update aerosol and atmospheric states in their Integrated
**Figure 18: Time series of the GOES-16 ABI Tb(10.35 μm) - Tb(12.3 μm) channel difference (˚C) highlighting**
**the morphology of dust over a seven-and-a-half hour time period. In addition, the time series demonstrates that**
**the dust could be tracked at night.**
Forecast System (IFS) (Morcrette et al. 2008; Benedetti et al. 2009), and the Japan Meteorological Agency (JMA),
which runs the Model of Aerosol Species in the Global Atmosphere (MASINGAR; Tanaka and Chiba 2005) along



with a 2DVar data assimilation method to provide operational aerosol-dust forecast and analysis (Yumimoto et al.,
2018). In addition, the NASA Global Modeling and Assimilation Office (GMAO) also assimilates aerosols in their
Goddard Earth Observing System Version 5 (GEOS-5; Randles et al., 2017).
Two types of approaches for the assimilation of aerosol exist. As summarized in (Penny et al., 2017, Zupanski, 2017),
one type is a weakly coupled data assimilation (WCDA) and a second type is a strongly coupled data assimilation
(SCDA). In the WCDA framework, assimilation of aerosol is conducted independently of the assimilation of coupled
atmospheric component, although the resulting analysis is used to initialize a coupled aerosol-atmosphere forecast to
allow interaction between the two components (e.g., Sekiyama et al., 2010, Rubin et al., 2017). In contrast, in the
SCDA framework, assimilation of aerosol and atmospheric components is performed simultaneously, treating the
coupled aerosol-atmosphere system as a single integrated system (e.g., Liu et al., 2011, Lee et al., 2017).
Analysis and forecast of dust or aerosol distribution can benefit from the assimilation of satellite data that may contain
a dust or aerosol signal. Satellite data of aerosol generally fall into two categories: satellite derived aerosol retrieval
products and satellite radiances affected by aerosol. An example of the former category is satellite retrieved aerosol
optical depth (AOD; e.g., Hsu et al. 2006, Levy et al. 2013, Remer et al. 2013). Channel differencing of infrared
brightness temperatures, ABI Tb(10.35 µm) - Tb(12.3 µm), discussed in this manuscript belongs to the latter category.
For both the retrieved products and the satellite radiances, a forward operator is required for the assimilation of data
into an NWP model. In particular, an accurate and fast radiative transfer model is critical for enabling direct
assimilation of satellite radiances (Weng, 2007).
Currently, the NOAA National Centers for Environmental Prediction (NCEP) employs the NOAA Environmental
Modeling System (NEMS) Global Forecast System (GFS) Aerosol Component (NGAC) for global dust forecasting
(Lu et al., 2016a). Development of NGAC is a collaborative on-going effort between NCEP and NASA toward aerosol
data assimilation capability in NCEP. Currently, there are plans to implement the aerosol assimilation capabilities in
Gridpoint Statistical Interpolation (GSI; Pagowski et al. 2014) and the Community Radiative Transfer Model (CRTM;
Han et al. 2006) into NGAC to allow the direct assimilation of satellite radiances affected by aerosol as well as
assimilation of AOD (Lu et al., 2016b). With that, the simple channel difference discussed in this manuscript can be
used to aid operational forecast of dust via data assimilation.
**5 National Weather Service Forecaster Perspective**
SAL airborne dust plumes can be transported across the Atlantic Ocean, along the southern periphery of a North
Atlantic subtropical high pressure system, towards South Florida. Transport of a SAL may be enhanced when a
subtropical high pressure system becomes zonally elongated towards southern portions of the United States, allowing
for more direct westward transport of a SAL towards South Florida. Thus, both detection and tracking of a SAL is
important for the preparation of potential impacts to South Florida.



As mentioned above, the scarcity of both in situ surface and upper-air observations, space-borne instruments are
essential across the tropical and subtropical Atlantic, Caribbean, and Gulf of Mexico basins. An important benefit of
having a space-borne sensor observe a SAL is the tendency of a SAL to transition from a relatively large homogeneous
air mass near Africa to an increasingly fractured, irregularly shaped dust plume during the westward migration across
the Atlantic basin. In addition, the appearance of SALs may also change due to encounters with dust-scavenging rain
systems of varying scale. Accurate and timely observations of smaller, irregularly shaped dust plumes, via products
derived from both geostationary and polar-orbiting satellites, is essential for anticipating important changes in lapse
rates and convective instability. Accurately discerning the horizontal and vertical extent of a SAL can aid the
prediction of severe weather potential. That is, National Weather Service (NWS) meteorologists of southern Florida
benefit greatly by tracking a SAL as it can be a proxy for the movement and evolution of an elevated mixed layer
(EML; Carlson and Ludlam 1968; Lanicci and Warner 1991). EMLs may lead to a dramatic increase in convective
available potential energy, especially when a SAL surmounts a maritime tropical air mass with high values of moist
static energy near the Florida peninsula.
Being that the mission of the National Weather Service is to provide forecasts and warnings for the protection of life
and property, it is vital that NWS meteorologists maintain situational awareness of the development and evolution of
a SAL and associated EML as both can quickly jeopardize this mission if overlooked. That is, knowledge of SAL
allows NWS meteorologists over South Florida to perform a better diagnosis of the mesoscale environment prior to
the onset of cumulus convection. Lower and middle-tropospheric thermodynamic structures, which accompany a SAL,
may often be missed by forecast models. An important consequence is for operational meteorologists to utilize the
latest observational capabilities that allow for the identification and tracking of suspended mixed-layer dust plumes.
Although a SAL has been known to influence local weather, a SAL also has the ability to impact both air quality and
visibility. Airborne dust can affect the health and safety of the public, either via direct respiratory impacts or indirectly
via reductions in horizontal surface visibility (Kuciauskas et al. 2018). In particular, decreases in the line-of-site
visibility has a direct impact on aviation. Furthermore, awareness of A SAL directly enhances the impact decision
support service (IDSS); IDSS is provided to core partners who rely on the NWS for timely and accurate severe weather
threat assessments in order to protect life and property.
**6 Summary and Conclusions**
This manuscript examined satellite observations of a SAL dust plume that moved from western Africa westward over
the eastern Atlantic Ocean. Observations from several sensors aboard satellite platforms were used herein: ABI
onboard GOES-16, VIIRS onboard NOAA-20, and CALIOP onboard CALIPSO. Further, the quantification of
vertically integrated water vapor was retrieved from two remote sources, each of which used multiple sensors from
multiple satellite: NUCAPS and MiRS. Satellite observations of the SAL dust plume presented herein extended from
16 February 2020 to 17 February 2020. Examination of the SAL dust plume employed GeoColor, low-level water
vapor, and split window difference imagery from GOES-16 ABI; True-Color and split window difference imagery



from VIIRS; VFM from CALIOP; gridded TPW and retrieved skew-t's from NUCAPS, and the ALPW product from
MiRS. Observational data from all of the aforementioned satellite platforms were used for the purpose of extending
and supporting two previous numerical studies, which hypothesized that water vapor may mask infrared detection of
dust.
Numerical studies have been used to examine the impact of water vapor on dust detection. Both Miller et al. (2019)
and Banks et al. (2019) used numerical methods to show that when vertically integrated water vapor increased above
some value, dust may be masked by water vapor; thus, making dust detection with simulated/synthetic infrared
imagery a challenge. Satellite observations of the African dust plume from 16-17 February 2020 provided
observational support for the two numerical studies just stated. Specifically, GeoColor imagery from ABI, True-Color
imagery from VIIRS, and the VFM from CALIOP all revealed the existence of dust in both the NDR and SDR.
However, both the $Tb(10.35\,\mu m) - Tb(12.30\,\mu m)$ split window difference from ABI and the EUMETSAT infrared
dust product suggested the existence of dust only in the NDR. Values of integrated water vapor exhibited a noticeable
difference between the NDR and the SDR.
Data from MiRS and NUCAPS are summarized presently. Specifically, at 0300 UTC 17 February 2020 values of the
ALPW product in the surface to 850 hPa layer decreased from the SDR, with values approximately 27.9 mm, to the
NDR, with values near 15.4 mm, an approximate 44 % decrease (Fig. 12). Further, values of the ALPW product in
the 700 hPa to 500 hPa layer decreased from the SDR, with values near 12.7 mm, to the NDR, with values near 2.5
mm, an approximate 80 % decrease (Fig. 13). In addition, at 0333 UTC 17 February 2020 values of gridded TPW
decreased from the SDR, with values near 26 mm, to the NDR, with values near 12 mm to 16 mm; an approximate 38
% decrease (Fig. 15). In both cases a distinct horizontal gradient of values of both the ALPW product from MiRS and
gridded TPW from NUCAPS existed near 15˚ N, the approximate boundary between the SDR and NDR. Furthermore,
the location of the distinct horizontal gradient of values of both the ALPW product and TPW were approximately co-
located with a distinct horizontal gradient of values of the $Tb(10.35\,\mu m) - Tb(12.30\,\mu m)$ split window difference from
ABI (Fig. 5) and the southern boundary of the dust signal in the EUMETSAT dust product (Fig. 6). That is,
observations show that dust within the SDR/NDR was masked/detected where values of both ALPW and gridded
TPW were the largest (sfc-805 hPa ALPW ~27.9 mm, TPW ~26 mm) /smallest (sfc-850 hPa ALPW ~15.4 mm, TPW
~14 mm). Furthermore, representative vertical sounding from NUCAPS exhibited a distinctly drier atmosphere in the
NDR compared to the SDR (Fig. 17). Consequently, satellite imagery and products of 16-17 February 2020 of an
African dust plume lend observational support to the numerical results of both Miller et al. (2019) and Banks et al.

32  (2019).

An important consequence of the observational study in this manuscript is relevant to NWS forecasters. There are two
aspects of a SAL that are important to NWS forecasters: (1) Dust in the SAL and (2) the associated EML within the
SAL. Dust within a SAL may impact not only respiratory function in people, but also aviation operations. An
associated EML within a SAL may lead to the development of severe thunderstorms. As a result the detection and



tracing of dust layers from Africa is important to NWS forecasters. Assimilation of dust in to operational forecast
models may help improve the forecasting of not only dust itself, but also the thermodynamic profile of an associated
EML with a SAL.
**Author Contributions**
LG and DB conceived the study of this paper, LG prepared the manuscript with the help of all co-authors. JT, JD, HC,
provided support for the acquisition of satellite data; JF and EB provided support with retrievals from MiRS and
NUCAPS; T.-C. W contributed data assimilation expertise; both HW and KK served as NWS collaborators, and SM
provided support for results from the MURI.
**Code/Data Availability**
AWIPS-2 code is protected by license and unavailable. Satellite data is available via the CLASS site.
**Competing Interests**
There are no competing interests.
**Acknowledgments**
The authors gratefully acknowledge that this research was primarily funded by the NOAA GOES-R Program Office
(NA19OAR4320073) along with the DOD/ONR/MURI (N00014-16-1-2040). The views, opinions, and findings in
this report are those of the authors, and should not be construed as an official NOAA and or U.S. Government position,
policy or decision.

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
