# Peer review of "Satellite Imagery and Products of the 16-17 February 2020 Saharan Air Layer"

_Atmospheric Measurement Techniques, 2020_

## Referee Comment (RC1) · Anonymous Referee #1 · 25 Oct 2020

This study examined the detection of dust aerosols in a Saharan Air Layer using ABI, VIIRS, and CALIOP products during a dust event in Feb. It is found that channel differences can largely detect dust in the northern region over the eastern Atlantic where water vapor is relatively low but cannot detect the mixture of smoke and dust in the southern region, where total precipitable water is high. The finding provides observational evidence to support recent modeling studies of water vapor masking the infrared detection of dust. The paper is overall well written with detailed information about infrared dust detection and thorough analysis. I only have a few suggestions for the

authors to consider before publication.

Specific comments:

1. While it is nice to add discussions about the implication of dust detection techniques in aerosol assimilation and weather forecast (sections 4-5), some details (although quite informative) are not very relevant to this study (e.g., lines 5-11, page 23). I suggest shortening these parts to keep the paper concise.

2. The paper discussed a lot of detailed aspects of infrared detection of dust, while most of them are very useful I think some content is not the main focus of this study and can be cut down. For instance, the discussion about whether air temperatures can explain the lateral changes in Tbs can be shortened (lines 19-24, page 13, lines 6-17, page 14), since the main hypothesis is that total precipitable water plays a major role.

3. Can you add some discussion about whether smoke may affect the detection of dust in the south region (along the black dashed line) in channel difference in Fig. 5?

Minor comments:

1. Which color indicates the negative value in Fig. 7a, light blue?

2. Is it possible to enlarge the font size of labels in Fig. 17? Can you add the information about green, white, red contours in the figure caption?

3. Lines 16 and 18, acronyms NRD and SRD are used but later NDR and SDR are used.

---

## Referee Comment (RC2) · Anonymous Referee #2 · 26 Nov 2020

Evaluation of overall quality The authors present a case study of a Saharan-like, dry dust airmass on 16-17 February 2020 to investigate whether higher water vapor concentrations can mask dust detections derived from satellite-based differential brightness temperature data as shown in numerical model-based studies of Miller et al. (2019) and Banks et al. (2019). For their analysis, the dry, dusty air mass off the west African coast is segregated into two partitions based upon the "dust" and "polluted dust" CALISPO vertical feature mask categorization. However, I feel that this segregation brings up the weakest element of this manuscript: the role of the polluted

air. In short, the authors draw much attention to polluted air early in the manuscript, but then drop all mention (or relevance) of the polluted airmass and any role it may play later in the manuscript to focus instead entirely on water vapor. After this initial stumble, the investigation gets back on track to its hypothesis that higher water vapor content can mask dust detections. The material presented afterwards, definitively proves, with the combination of GOES and VIRS visible and infrared imagery, that elevated dust present in visible satellite imagery south of 10N was clearly missed by differential brightness temperature bands often used for dust detection where GFS total precipitable water values exceeded 20 mm. They also demonstrate that the presence and amount of water vapor can be derived from satellite-based estimates of total precipitable water and the more recently developed advected total precipitable water data products. The other problematic part of the manuscript lies in sections 4 and 5, which I feel distract from the overall narrative. These sections highlight current aerosol data assimilation and importance of aerosols to weather forecasting in South Florida, but the discussion on these topics is severely limited by the lack of any concrete examples or examples of what and how differential brightness temperatures could be used to initialize aerosol data assimilation. Finally, once the authors show that dust is masked by higher water vapor content, the existence of this missed dust is largely forgotten and I am left wondering as to how significant this problem is or if any methods may exist to detect this masked dust. Despite, these reservations about some elements of the manuscript, I do feel that the investigation presented does meet is investigative goals and provides a clear and relevant example of how water vapor can does mask dust detections from commonly used satellite-based datasets and that this work should be published following some major revisions.

Specific Comments/Questions

Major:

1) In section 3, much emphasis is given the presence of polluted air within the ubiquitous presence of Saharan dust in the SDR as compared to the more pristine NDR.

ubiquitous

After this section, the presence of polluted air is not mentioned again because of the paper's emphasis on masked dust detections in regions of higher water vapor content. Do you believe that the presence of polluted air has any discernable impact upon the ability differential Tb methods to detect dust layers or the on the values of differential Tb (i.e., more negative, less negative)?

2) I feel that sections 4 and 5 are almost an entirely different paper because material from these sections feel disconnected from the narrative. Material from these sections is not referenced in the abstract or as a key goal of this investigation. I do not dispute the importance of dust data assimilation for weather forecasting and health, but neither section shows or cites a concrete example of an application of differential brightness temperature dust detection being applied to improve data assimilation or weather forecasting. In short, something to help illustrate how you are improving over existing techniques would be quite helpful to tie everything together.

3) How significant is the problem is missed dust detections due to water vapor? The manuscript successfully shows dust is indeed masked by higher vapor concentrations, but what is the impact from this result? Should we be concerned that so much dust is missed? Are there any potential methods or suggested methods to address this issue?

Minor:

4) I would strongly advise against using the terms "SAL dust or SAL transport". Although dust is often associated with the SAL, dust is neither exclusive to it, nor is it always present within it. Figures 2 and 17, help illustrate this point where your sounding profiles (Figure 17) show the SAL to be elevated above the low-level marine layer and below the free troposphere, yet in Figure 2, the CALIPSO VFM shows dust being present from 3.0 kms to the surface. Furthermore, your paper does not limit itself to just results from just within the SAL.

5) Page 16, TPW and TB differencing. You show that you can successfully discriminating between the NDR and SDR regions based upon TPW and differential Tbs for

your case study. Aside from the critical TPW value (i.e., Miller et al.) are there any other limitations to this method? Would you expect the nature of the aerosol mixture to have any impact should polluted air mix with the more pristine dust environment of the NDR?

6) Page 17, Figs. 12 and 13, ALPW: In Gitro et al. 2018, ALPW data products are noted to have 3-hourly output (including for 1800 UTC) because it leverages GFS analysis winds to advect LPW fields to a common analysis time despite its dependence upon LEO satellite data. In your manuscript, you show data for 0300 UTC 17 Feb. 2020, which you rationalize while stating, "Due to the use of LEO sensors, imagery for the ALPW was not necessarily available as often as ABI data from GOES-16. Subsequently, retrieved values of ALPW, in the layer from the surface to 850 hPa, valid at 0300 UTC 17 February 4 2020 are displayed in Fig. 12". In light of Gitro et al. 2018 and your own explanation not being commensurate, is there another reason why ALPW data at 1800 UTC itself was not used and then compared to Figures 8-11, especially because exact ALPW values are cited. I do agree with the author's assertion that dust layers are slowly evolving features (i.e., Fig. 18). The time difference seems unnecessary, even if your rationalization is use it as reference for the NUCAPS data, which are dependent upon the later overpass times (0333 UTC) of CrIS and ATMS instruments, and your dust airmasses of NDR and SDR are still distinct and slowly evolving.

7) Did you find that the critical vertical integrated water vapor values presented by Miller et al. (2019) and Banks et al. (2019) for dust masking by water vapor were consistent with your results? I would imagine that NWS forecasters might have a strong interest in knowing how well constrained the potential critical values are when applying the differential Tbs technique to find pockets of elevated, dusty air.

8) Would you expect the efficacy of differential Tb techniques to have any seasonal dependence?

Technical Corrections

[Figure]

General:

1) Line numbering. For review, line numbers should be continuous for the entire manuscript and not reset on each page

2) Spacing following periods. I see examples of both one and two spaces in the manuscript. Please choose one method and then make the manuscript consistent.

3) Numerous typos. While it did not prevent me from understanding your paper, they were enough to be distracting. Please check for these typos with fresh eyes.

4) Incorrect usage of "Since" and "Because". Since implies the passage of time (i.e., Since 1972) where because implies a reason for something.

5) Figure colorbars. For figures using GOES-16 data the color bar label text values tend to be rather small and hard to read. Can the font be modified or made larger?

6) Tendency for over usage of prepositions. Throughout the manuscript, there were numerous instances of phrases such as "...values of the difference between the temperature and dewpoint temperature..." or "...of values of TPW...". While not grammatically incorrect, this type of phasing comes off as excessively wordy and harder to read. Instead these phrases could be re-written as, "...difference in dewpoint and temperature values..." or "of TPW values".

7) By definition the SAL itself is strictly defined a region of dry, well-mixed air that is bounded by the maritime layer below and free troposphere above where temperatures and warm and the air is dry due to dry, convective mixing upstream in the Sahara. Therefore it is somewhat taboo to say the "SAL is transported" or the "SAL is fractured". Furthermore, a SAL is not required to contain dust (even though it is often associated with it). Instead for this section I might rephrase to say "transport of dust associated with a Saharan-like, elevated mixed layer is" because what you are tracking from satellite are the region of dusty, dry, well-mixed air mass westward across the Atlantic, which are eroded by airmass intrusions as it progresses across the Atlantic. This comment is

mainly associated with section 5.

8) Given 7, a SAL present over the Atlantic is by definition an EML. I would advise picking and using only one term or using a term such as "Saharan-like EML". Switch between these terms is most common in section 5 and the end of page 25.

9) Typo on Page 26, line 1: should read as "into" rather than "in to"

Specific:

1) Typo on Page 2, line 33: should read as "...from two sources..."

2) Grammar error on Page 2, line 34: should read as "...data were..." rather than "...data was..."

3) Typo on Page 3, line 15: remove the extra space between 'which' and 'are'

4) Typo on Page 3, line 20: should read "... was launched on..."

5) Typo on Page 3, line 21: remove the comma after operational

6) Typo on Page 3, line 24: "one-half orbit" should read as "half an orbit"

7) Define term on Page 3, line 30: Explicitly define low-earth orbiting as LEO. The acronym is used later without having be defined first here.

8) Typo on Page 3, line 31: remove the comma after "...2009)"

9) Clarification on Page 3, line 31: not sure of your intent in the phrase "A component of CALIOP is a lidar" because CALIOP itself is the lidar aboard CALIPSO. Please clarify your meaning.

10) Typo on Page 4, line 14: remove "tropical" before 'Saharan Air Layer' for consistency.

11) Typo on Page 4, line 17: should read as "...end-user feedback..."

12) Typo on Page 4, line 19: add "its" between "enable" and "use"

13) Typo on Page 4, line 20: Add a hyphen after "0.5"

14) Typo on Page 4, line 34: should be "16-km footprint"

15) Define term on Page 4, linear 27: CIRA used 3 times in paper, but is not defined.

16) Typo on Page 3, line 36: Replace "Since" with "Because"

17) Type on Page 5, lines 8 and 9: add hyphen before "km"

18) Type on Page 5, line 15: replace "Since" with "Because"

19) Typo on Page 5, line 16: should read as zero-hour forecasts

20) Typo on Page 5, line 18: remove command after "(Fig. 1)"

21) Suggested edit, Page 5, line 19: Latter half of sentence reads oddly to me. I would suggest revising to "….contour, located over the eastern Atlantic."

22) Minor correction, Page 6, Figure 2: the text of manuscript makes direct reference to Figure 2a and Figure 2b on page 7. It is understood from context, but I suggest either modifying the figure caption to include 2a and 2b or add an 'a' and 'b' on your figure panels.

23) Typo on Page 7, line 1: should read "…an ascending CALIPSO overpass…" because there is only one overpass in reference to Fig. 2.

24) Typo on Page 7, line 9: add "the" before first "VFM"

25) Suggestion on Page 7, lines 8-11: I believe your statement is not quite correct. All the VFM 'sees' is that the polarization signal happens to cross the threshold criteria between 'dust' and 'polluted dust'. While it does separate regions of more significant pollutant concentrations, it does not necessarily mean that notable pollutant concentrations are not present in regions noted as being 'dust'. It is just below the threshold to flag it as polluted dust. I would suggest revising to emphasize that dust was ubiquitous for the entire transect, but the greatest pollution concentrations are found south of 15N.

26) Suggested revision on Page 7, lines 13-18: I would revise this paragraph. A hypothesis assumes that you will either validate or refute it in your study, but you are only stating your assumptions for what will be characterized for the NRD and SRD. My suggestion would be to remove the paragraph and just add a one sentence to the end of the last one saying "For this study, we assume regions along the CALIPSO transect northward of 15N were only dust and regions south of 10N. . ..."

27) Suggestion on Page 8, line 2: I would sell that this thickness range is consistent with Figure 2.

28) Suggested a revision on Page 8, line 2-4: Reads awkwardly, please revise. I would suggest combining the statement (line 2-3) with the question (lines 2-4).

29) Typo on Page 8, line 19: replace "diagnosed" with "derived"

30) Suggestion on Pages 8 and 9: Your caption in Figure 4 point to Figure 5, which comes after it, which is not ideal. Because Figures 4 and 5 are so closely related, you may want to consider merging the two figures together as Figure 4a and 4b. This your annotations make more sense because you can directly see what is being emphasized without the need for additional explanation.

31) Suggested revision on Page 9, line 4: replace start with "Specifically, difference plots of brightness temperature values (Tbs) at 12.3. . ..". The original reads oddly with all the "of's".

32) Missing information on Page 9 lines 6 and 7: Although you are pointing to Miller et al. (2019), it would be more helpful if you stated what the critical value of vertical integrated water vapor is, otherwise your meaning comes off as rather vague. Is the content high? Low?

33) Removal on Page 8, lines 8 – 12: After "Although", the information is redundant and does not add anything to your manuscript.

34) Removal on Page 8, Line 21-22: I would suggest removing sentence starting "With

that" because it just repeats the same statement as the sentence coming before it. Instead skip right to the example.

35) Add detail on Page 8, Line 4-5: As this is a new paragraph, I suggest adding the references to Figures 4 and 5 in parentheses after "channel difference image" and "GeoColor image".

36) Typo on Page 10, line 6: should read as "...upper-left portion..."

37) Suggested revision on Page 10, line 6: The white letter "A" is on Figure 4, yet you are talking about Figure 5. I would either mention that it is in Figure 4 or just say "denoted by the letter "A".

38) Suggested merger on Page 11, line 12 and 13: Reads a little choppy, I would suggest merging the "Note" and that "That is" sentences.

39) Revision or removal of sentence on Page 11, line 14-15. You clear illustrate that the TBs difference method for dust detection is not foolproof, but the latter part of the sentence reads a bit too vague because it is not clear as to what the "other components" refers to. Will you talk about them? Are you referencing other work? Some clarity here would help.

40) Page 12, line 15: Would be helpful to know what this critical integrated water vapor value is. 41) Typo on Page 12, line 20: "Since" should be "Because"

42) Suggested modification on Page 13, line 13: Larger value can imply either larger positive or negative values. I would replacing "larger" with "positive" to remove any ambiguity.

43) Suggestion on Page 13, line 23: Remove the "stated differently" phrase. You say the Tbs difference is 15C, yet -18C and -2C are 16C apart. Is this a typo or a rounding problem? When combined with the additional directions, it makes this statement more muddled.

44) Suggested correction on Page 13, line 24: replace "...in values of Tbs of the..." with "in Tbs values of the...". To many prepositions here reads oddly.

45) Typo on Page 14, line 6: "Since" should be "Because"

46) Grammar error on Page 14, line 17: The colon use within the sentence makes no grammatical sense. Did you intend for a semi-colon? Please revise.

47) Suggested correction on Page 14, line 19: Need a better transition. You have the what "Values of TPW are shown in relation to various satellite fields". This sentence is grammatically correct, but it is a statement without the context. You give me a statement, but not a motivation on what you will do. "I am doing X to investigate Y."

48) Minor correction on Page 16, line 5: "Largest" can be both positive or negative, it would be unambiguous if either "warmest" or "highest" were used instead.

49) Suggested edit on Page 16, line 15: Although you mention it in your caption, it would be worthwhile to also mention the channels being differenced are 10.35 and 12.3 microns.

50) Suggested revision on Page 16, line 15-18: Sentence starting with "Not only" is quite wordy and hard to follow. I would suggest revising it into a more concise form such as "These figures show the NDR to be co-located with a, b, and c."

51) Sentence too vague or out of place on Page 16, lines 19-20: The last sentence is terribly vague and just seems out of place. Which sensors? What is the significance?

52) Suggestion on Page 18, line 7: You mention that ALPW decreases from south to north and I can see this in your data. Perhaps it might be worth adding an annotation to Figures 12 to show exact where the 27.9 mm and 15.4 mm ALPW values are being estimated.

53) Suggested replacement on Page 19, line 2. Replace "retrieval of values of ALPW" with "retrieval fo ALPW values" to remove excessive prepositions.

[Figure]

54) Possible logic error on Page 19, lines 2-3: The sentence has circular logic because "values of TPW are show in Fig. 15", but then later in the same sentence say it "was the time of the granule show in Fig. 15". Do you mean to reference a different figure?

55) Suggestion on Figs 12, 13, and 14: You often cite 15N because it divides your pristine NDR and polluted SDR environments. I think it would be useful to consider annotating a line to each of these figures to show the 15N parallel for reference, especially for figure 14 where the coastline is muddied by the show Tb difference data.

56) Overly wordy phrase on Page 20, line 6: "values of the difference between the temperature and dewpoint temperature". I would consider revising to be more concise.

57) Suggestion on Page 20, line 19: Instead of using the phrase "appeared earlier in the manuscript", I would consider adding "(Fig. 10)" after "images" on line 18 to make your wording more exact.

58) Typo on Page 23, line 7: should read as "...assimilation of aerosols..."

59) Awkward working on Page 23, line 7, "...assimilation of the coupled atmospheric component...". It is no clear as to what the atmosphere is coupled, I would suggest removing "coupled" to make the distinction between WCDA and SCDA more clear.

60) Typo on Page 23, line 14: should read as "...data of aerosols..."

61) Typo on Page 23, line 19: should read as "...a NWP model."

62) Logic error on Page 24, line 17: Once over the Atlantic, the SAL is by definition an EML.

63) Typo on Page 24, line 36: should read as "...split-window difference..."

64) Additional detail needed Page 25, line 8: It would be useful to know what this "some value" is.

65) Typo on Page 25, line 20 and line: remove the space before "

66) Typo on Page 27, line 17: Remove the hyperlink associated with the doi

---

## Author Comment (AC1) · 14 Dec 2020

Date: 14 December 2020

Purpose: Reply to referee 1

Uploaded Document Name: LGrasso_African_Dust_14_Dec_2020_reply_to_referee1.pdf

Specific comments:

1. While it is nice to add discussions about the implication of dust detection techniques in aerosol assimilation and weather forecast (sections 4-5), some details (although quite informative) are not very relevant to this study (e.g., lines 5-11, page 23). I suggest shortening these parts to keep the paper concise.

I removed the following two paragraphs along with stated references from the assimilation section:

[revised manuscript text omitted]

2. The paper discussed a lot of detailed aspects of infrared detection of dust, while most of them are very useful I think some content is not the main focus of this study and can be cut down. For instance, the discussion about whether air temperatures can explain the lateral changes in Tbs can be shortened (lines 19-24, page 13, lines 6-17, page 14), since the main hypothesis is that total precipitable water plays a major role.

I have to disagree. Values of Tbs for water vapor imagery are influenced by both water vapor and temperature. My goal with the discussion is to demonstrate to a reader that horizontal variations of temperature are unable to explain horizontal variations of values of Tbs of water vapor imagery. Consequently, the text focuses on horizontal variations of water vapor to explain horizontal variations of values of Tbs.

3. Can you add some discussion about whether smoke may affect the detection of dust in the south region (along the black dashed line) in channel difference in Fig. 5?

Yes, such a discussion can only benefit a reader. Consequently, the following text was as the fourth paragraph after Fig. 7:

In addition to dust, smoke from biomass burning over Africa (Fig. 3) existed within the SDR. One open question is that smoke may impact values of Tb(10.35 µm) - Tb(12.3 µm) in such a way as to mask the dust in the SDR. Based on previous satellite observations, Hillger and Ellrod (2003) have shown that if a layer of smoke is optically thick enough in infrared bands, Tbs of smoke will appear cool. However, cool Tbs associated with smoke may be confused with cool, elevated, land surfaces. Further, smoke layers were undetected in values of infrared channel differences.  As part of a discussion of the utility of the day-night band on the VIIRS sensor, Miller et al. (2013) also point out the inability of smoke detection by infrared satellite imagery. One consequence of these two studies suggests that smoke within the SDR, was unable to mask dust in the SDR. Another mechanism for dust masking in the SDR is sought.

Minor comments:

1. Which color indicates the negative value in Fig. 7a, light blue?
Fig. 7a is a True-Color image from three reflective bands from VIIRS. As such, there are no negative values. Perhaps you meant Fig. 7b; in this case, the color table for Fig. 7b is the same as that in Fig. 5. Negative values are light blue.

2. Is it possible to enlarge the font size of labels in Fig. 17? Can you add the information about green, white, red contours in the figure caption?

Fig. 17 has been cleaned up a bit. Extra lines that represented a lifted parcel have been removed as the temperature and dewpoint temperature difference is the main focus of both NUCAPS soundings. Pressure level values were increased as were the mixing ratio and temperature values on the horizontal axis. Now one can focus on the T-Td spread in sounding 1 and 2.

3. Lines 16 and 18, acronyms NRD and SRD are used but later NDR and SDR are used.

You have eyes of an Eagle: I changed NRD and SRD to NDR and SDR; respectively.

Kind Regards,

Louie

---

## Author Comment (AC2) · 14 Dec 2020

Date: 14 December 2020

Purpose: Reply to referee 2

Uploaded Document Name: LGrasso_African_Dust_14_Dec_2020_reply_to_referee2.pdf

Evaluation of overall quality The authors present a case study of a Saharan-like, dry dust airmass on 16-17 February 2020 to investigate whether higher water vapor concentrations can mask dust detections derived from satellite-based differential brightness temperature data as shown in numerical model-based studies of Miller et al. (2019) and Banks et al. (2019). For their analysis, the dry, dusty air mass off the west African coast is segregated into two partitions based upon the "dust" and "polluted dust" CALISPO vertical feature mask categorization. However, I feel that this segregation brings up the weakest element of this manuscript: the role of the polluted air. In short, the authors draw much attention to polluted air early in the manuscript, but then drop all mention (or relevance) of the polluted airmass and any role it may play later in the manuscript to focus instead entirely on water vapor. After this initial stumble, the investigation gets back on track to its hypothesis that higher water vapor content can mask dust detections. The material presented afterwards, definitively proves, with the combination of GOES and VIRS visible and infrared imagery, that elevated dust present in visible satellite imagery south of 10N was clearly missed by differential brightness temperature bands often used for dust detection where GFS total precipitable water values exceeded 20 mm. They also demonstrate that the presence and amount of water vapor can be derived from satellite-based estimates of total precipitable water and the more recently developed advected total precipitable water data products. The other problematic part of the manuscript lies in sections 4 and 5, which I feel distract from the overall narrative. These sections highlight current aerosol data assimilation and importance of aerosols to weather forecasting in South Florida, but the discussion on these topics is severely limited by the lack of any concrete examples or examples of what and how differential brightness temperatures could be used to initialize aerosol data assimilation. Finally, once the authors show that dust is masked by higher water vapor content, the existence of this missed dust is largely forgotten and I am left wondering as to how significant this problem is or if any methods may exist to detect this masked dust. Despite, these reservations about some elements of the manuscript, I do feel that the investigation presented does meet is investigative goals and provides a clear and relevant example of how water vapor can does mask dust detections from commonly used satellite-based datasets and that this work should be published following some major revisions.

Specific Comments/Questions

Major:

1) In section 3, much emphasis is given the presence of polluted air within the ubiquitous presence of Saharan dust in the SDR as compared to the more pristine NDR. After this section, the presence of polluted air is not mentioned again because of the paper's emphasis on masked dust detections in regions of higher water vapor content. Do you believe that the presence of polluted air has any discernable impact upon the ability differential Tb methods to detect dust layers or the on the values of differential Tb (i.e., more negative, less negative)?

In the manuscript, I provided evidence that smoke from biomass burning over African (Fig. 3) is the source of polluted air with the SDR. As such, I added the following text four paragraphs below Fig. 7 in an attempt to address your questions.

In addition to dust, smoke from biomass burning over Africa (Fig. 3) existed within the SDR. One open question is that smoke may impact values of Tb(10.35 μm) - Tb(12.3 μm) in such a way as to mask the dust in the SDR. Based on previous satellite observations, Hillger and Ellrod (2003) have shown that if a layer of smoke is optically thick enough in infrared bands, Tbs of smoke will appear cool. However, cool Tbs associated with smoke may be confused with cool, elevated, land surfaces. Further, smoke layers were undetected in values of infrared channel differences.  As part of a discussion of the utility of the day-night band on the VIIRS sensor, Miller et al. (2013) also point out the inability of smoke detection by infrared satellite imagery. One consequence of these two studies suggests that smoke within the SDR, was unable to mask dust in the SDR. Another mechanism for dust masking in the SDR is sought.

2) I feel that sections 4 and 5 are almost an entirely different paper because material from these sections feel disconnected from the narrative. Material from these sections is not referenced in the abstract or as a key goal of this investigation. I do not dispute the importance of dust data assimilation for weather forecasting and health, but neither section shows or cites a concreate example of an application of differential brightness temperature dust detection being applied to improve data assimilation or weather forecasting. In short, something to help illustrate how you are improving over existing techniques would be quite helpful to tie everything together.

First, I have reduced the size of both sections 4 and 5 at the request of reviewer 1. Consequently, the reduction may help to lessen or remove the feel of disconnect from the main narrative. Section 4 contains a rather detailed discussion about dust assimilation at many forecasting centers. At the end of the section is stated, "…the simple channel difference discussed in this manuscript can be used to aid operational forecast of dust via data assimilation. " Inclusion of a concreate example is outside the scope of the manuscript and would, as you suggested, transform section 4 into the beginnings of a new paper. In section 5, a forecaster from South Florida tells a reader that, "both detection and tracking of a SAL is important for the preparation of potential impacts to South Florida." Further in the section, a reader is informed about a specific type of impact, "Accurately discerning the horizontal and vertical extent of a SAL can aid the prediction of severe weather potential". As for your request for a concreate example, NWS forecasters simply do not have the luxury of time to do so. When I invited NWS forecasters to participate in this paper, one of the first concerns conveyed to me as that they have limited time. My goal here was to give NWS forecasters an opportunity to speak directly to a reader. As for my illustrating how I am improving over existing techniques, I claim this paper does just that; provide observational support to numerical studies to advance our understanding of dust detection.

3) How significant is the problem is missed dust detections due to water vapor? The manuscript successfully shows dust is indeed masked by higher vapor concentrations, but what is the impact from this result? Should we be concerned that so much dust is missed? Are there any potential methods or suggested methods to address this issue?

All questions were addressed in the following text, which was added at the end of Section 3.

As shown above, detection of dust in the SDR, by means of the infrared channel difference, was masked by water vapor. Undetected dust layers may hamper studies of both the direct radiative effect–scattering of energy by dust particles–and the indirect radiative effect–microphysical impacts on cloud lifetimes. Further, undetected dust layers may pose a hazard to both civilian and military aviation through a reduction of visibility and potential damage to aircraft engines. Undetected dust presents a different significance and concern depending on the application. For example, hazard to aircraft may be deemed more significant and a higher level of concern compared to scattering of solar and longwave radiation by undetected dust layers. As discussed above, GeoColor imagery and the CALIOP instrument on CALIPSO detected dust in the SDR. However, both methods relied on measurements of reflected solar energy; as a result, dust will go undetected after sunset in the SDR. However, dust in the NDR was not only detected, but also tracked after sunset. One potential method for nighttime detection of dust in the SDR may come from a future day-night band (DNB, Miller et al. (2013)) on a geostationary satellite. Nighttime dust and smoke detection may be afforded by a DNB thought the measurement of reflected moonlight.

Minor:

4) I would strongly advise against using the terms "SAL dust or SAL transport". Although dust is often associated with the SAL, dust is neither exclusive to it, nor is it always present within it. Figures 2 and 17, help illustrate this point where your sounding profiles (Figure 17) show the SAL to be elevated above the low-level marine layer and below the free troposphere, yet in Figure 2, the CALIPSO VFM shows dust being present from 3.0 kms to the surface. Furthermore, your paper does not limit itself to just results from just within the SAL.

A nice distinction between "SAL dust" and "dust associated with a SAL"; thus all instances of "SAL dust" were replaced with "dust associated with a SAL". However, "SAL transport" or "transport of a SAL" is another matter, which is addressed below.

5) Page 16, TPW and TB differencing. You show that you can successfully discriminating between the NDR and SDR regions based upon TPW and differential Tbs for your case study. Aside from the critical TPW value (i.e., Miller et al.) are there any other limitations to this method? Would you expect the nature of the aerosol mixture to have any impact should polluted air mix with the more pristine dust environment of the NDR?

Aside from critical values of TPW, I am unable to think of any limitations.

I hesitate to provide an answer as you are asking for my expectation.  I will however, provide a hesitant reply anyway. Our experience with satellite data suggests that pollution from human activities and/or smoke from biomass burning goes undetected in longwave channel differencing.

6) Page 17, Figs. 12 and 13, ALPW: In Gitro et al. 2018, ALPW data products are noted to have 3-hourly output (including for 1800 UTC) because it leverages GFS analysis winds to advect LPW fields to a

common analysis time despite its dependence upon LEO satellite data. In your manuscript, you show data for 0300 UTC 17 Feb. 2020, which you rationalize while stating, "Due to the use of LEO sensors, imagery for the ALPW was not necessarily available as often as ABI data from GOES-16. Subsequently, retrieved values of ALPW, in the layer from the surface to 850 hPa, valid at 0300 UTC 17 February 4 2020 are displayed in Fig. 12". In light of Gitro et al. 2018 and your own explanation not being commensurate, is there another reason why ALPW data at 1800 UTC itself was not used and then compared to Figures 8-11, especially because exact ALPW values are cited. I do agree with the author's assertion that dust layers are slowly evolving features (i.e., Fig. 18). The time difference seems unnecessary, even if your rationalization is use it as reference for the NUCAPS data, which are dependent upon the later overpass times (0333 UTC) of CrIS and ATMS instruments, and your dust airmasses of NDR and SDR are still distinct and slowly evolving.

I see your point, a very good one. I have no idea why I used a 0300 ALPW field to support the distribution of water vapor at 1800 UTC. I would bring up the same point you did if I saw someone else do what I did. Here is what I think I was thinking at the time, maybe…: When I first saw the plots of ALPW at the different levels, I likely, perhaps, said to myself, "Hey, the horizontal distribution of values of ALPW is similar to the horizontal values of TPW in the figures I was just thinking and writing about." I may have been encouraged that all the piece were coming together; you know, caught up in the joy of the moment. If the morphology of the event was faster, then I would fix the text in this manuscript. Actually, now that I think about it, I never would have used 0300 UTC ALPW to support values of TPW from nine hours earlier; there would be a noticeable difference in the horizontal patters due to speedier evolution.

7) Did you find that the critical vertical integrated water vapor values presented by Miller et al. (2019) and Banks et al. (2019) for dust masking by water vapor were consistent with your results? I would imagine that NWS forecasters might have a strong interest in knowing how well constrained the potential critical values are when applying the differential Tbs technique to find pockets of elevated, dusty air.

I was tempted to speak about any possible consistency between the two studies you mention and this observational study; especially since I was the one who produced the synthetic imagery for the Miller et al. study. As tempting as it is to speak about the values in the previous work, I decided such information could be misunderstood and used incorrectly. For a given amount of TPW, one can easily reduce values of Tb(10.35um)-Tb(12.3um) from positive—missed dust, to negative—detected dust: Simply increase the depth of the dust layer. How do I know? That is exactly what I have done with other numerical simulations. In other words, there does not exists a one-to-one mapping of values of the channel and … wait, I've already typed this in the manuscript.

8) Would you expect the efficacy of differential Tb techniques to have any seasonal dependence?

As in my reply to your question 5) above, I hesitate to discuss expectations. I have not done any seasonal studies. My goal with this manuscript is to stay focused on providing observational support to the Miller et al. (2019) and Banks et al. (2019) studies.

Technical Corrections

General:

1) Line numbering. For review, line numbers should be continuous for the entire manuscript and not reset on each page

Because we are already with the review process, I'll keep things the way they are.

2) Spacing following periods. I see examples of both one and two spaces in the manuscript. Please choose one method and then make the manuscript consistent.

I used Ctrl-F and hit the space bar twice and voila! I found not only a double space after periods, but also between some words within a sentence.

3) Numerous typos. While it did not prevent me from understanding your paper, they were enough to be distracting. Please check for these typos with fresh eyes.

I see below you list them. I'll address typos below.

4) Incorrect usage of "Since" and "Because". Since implies the passage of time (i.e., Since 1972) where because implies a reason for something.

Found nine instances of incorrect word usage and changed "since" to "because".

5) Figure colorbars. For figures using GOES-16 data the color bar label text values tend to be rather small and hard to read. Can the font be modified or made larger?

I also wondered the same thing and was told that a reader will be able to click an image and make it bigger. Thus, I left things as they are.

6) Tendency for over usage of prepositions. Throughout the manuscript, there were numerous instances of phrases such as ". . .values of the difference between the temperature and dewpoint temperature. . ." or ". . .of values of TPW. . .". While not grammatically incorrect, this type of phasing comes off as excessively wordy and harder to read. Instead these phrases could be re-written as, ". . .difference in dewpoint and temperature values. . ." or "of TPW values".

You're attacking my writing style. As you pointed out, with a double negative, that the grammar is correct. That you may find my usage of the prepositions excessively wordy is your opinion. I'm under the impression that my job is to produce a manuscript that is written well.

7) By definition the SAL itself is strictly defined a region of dry, well-mixed air that is bounded by the maritime layer below and free troposphere above where temperatures and warm and the air is dry due to dry, convective mixing upstream in the Sahara. Therefore it is somewhat taboo to say the "SAL is transported" or the "SAL is fractured". Furthermore, a SAL is not required to contain dust (even though it is often associated with it). Instead for this section I might rephrase to say "transport of dust associated with a Saharan-like, elevated mixed layer is" because what you are tracking from satellite are

the region of dusty, dry, well-mixed air mass westward across the Atlantic, which are eroded by airmass intrusions as it progresses across the Atlantic. This comment is mainly associated with section 5.

Back in 4) under "minor" I indicated that I changed "SAL dust" to "dust associated with a SAL" in the manuscript. However, the change excluded the Forecaster Perspective section. As indicated above, my purpose of inviting forecasters to participate in this paper was to give them a voice. I understand that the text is within a peer-reviewed manuscript; however, I must protect the forecaster's style of communication. If I change what they wrote to what you want, you will be censoring them.

8) Given 7, a SAL present over the Atlantic is by definition an EML. I would advise picking and using only one term or using a term such as "Saharan-like EML". Switch between these terms is most common in section 5 and the end of page 25.

See above.

9) Typo on Page 26, line 1: should read as "into" rather than "in to" fixed

Specific:

1) Typo on Page 2, line 33: should read as ". . .from two sources. . ." fixed

2) Grammar error on Page 2, line 34: should read as ". . .data were. . ." rather than ". . .data was. . ." fixed

3) Typo on Page 3, line 15: remove the extra space between 'which' and 'are' fixed

4) Typo on Page 3, line 20: should read ". . . was launched on. . ." fixed

5) Typo on Page 3, line 21: remove the comma after operational fixed

6) Typo on Page 3, line 24: "one-half orbit" should read as "half an orbit" fixed

7) Define term on Page 3, line 30: Explicitly define low-earth orbiting as LEO. The acronym is used later without having be defined first here. I'm confused, LEO is defined on page 1, line 24.

8) Typo on Page 3, line 31: remove the comma after ". . .2009)" fixed

9) Clarification on Page 3, line 31: not sure of your intent in the phrase "A component of CALIOP is a lidar" because CALIOP itself is the lidar aboard CALIPSO. Please clarify your meaning. Changed to "CALIOP, a lidar, emits packets…"

10) Typo on Page 4, line 14: remove "tropical" before 'Saharan Air Layer' for consistency. fixed

11) Typo on Page 4, line 17: should read as ". . .end-user feedback. . ." fixed

12) Typo on Page 4, line 19: add "its" between "enable" and "use" fixed

13) Typo on Page 4, line 20: Add a hyphen after "0.5" fixed

14) Typo on Page 4, line 34: should be "16-km footprint" fixed

15) Define term on Page 4, linear 27: CIRA used 3 times in paper, but is not defined. See page 1, line 7: Cooperative Institute for Research in the Atmosphere (CIRA)

16) Typo on Page 3, line 36: Replace "Since" with "Because" fixed

17) Type on Page 5, lines 8 and 9: add hyphen before "km" fixed

18) Type on Page 5, line 15: replace "Since" with "Because" fixed

19) Typo on Page 5, line 16: should read as zero-hour forecasts fixed

20) Typo on Page 5, line 18: remove command after "(Fig. 1)" fixed

21) Suggested edit, Page 5, line 19: Latter half of sentence reads oddly to me. I would suggest revising to ". . ..contour, located over the eastern Atlantic." fixed

22) Minor correction, Page 6, Figure 2: the text of manuscript makes direct reference to Figure 2a and Figure 2b on page 7. It is understood from context, but I suggest either modifying the figure caption to include 2a and 2b or add an 'a' and 'b' on your figure panels. 'a' and 'b' along with 'a)' and 'b)' were added to the figure panels and caption, respectively.

23) Typo on Page 7, line 1: should read ". . .an ascending CALIPSO overpass. . ." because there is only one overpass in reference to Fig. 2. fixed

24) Typo on Page 7, line 9: add "the" before first "VFM"

25) Suggestion on Page 7, lines 8-11: I believe your statement is not quite correct. All the VFM 'sees' is that the polarization signal happens to cross the threshold criteria between 'dust' and 'polluted dust'. While it does separate regions of more significant pollutant concentrations, it does not necessarily mean that notable pollutant concentrations are not present in regions noted as being 'dust'. It is just below the threshold to flag it as polluted dust. I would suggest revising to emphasize that dust was ubiquitous for the entire transect, but the greatest pollution concentrations are found south of 15N.

I disagree. Line 8 into 9 reads, "North of approximately 15˚ N, the VFM suggested dust was the primary constituent in the aerosol layer. " Note the grouping "primary constituent" implies there may be others constituents. Yet, I do like your phrasing so I included the following as the last line in the paragraph, "That is, dust was ubiquitous for the entire transect with the greatest pollution concentrations found south of 15˚ N.

26) Suggested revision on Page 7, lines 13-18: I would revise this paragraph. A hypothesis assumes that you will either validate or refute it in your study, but you are only stating your assumptions for what will be characterized for the NRD and SRD. My suggestion would be to remove the paragraph and just add a one sentence to the end of the last one saying "For this study, we assume regions along the CALIPSO transect northward of 15N were only dust and regions south of 10N. . ..."

Take a look at the text under Fig. 7 on page 11. There you'll see the following: "One of the assumptions stated above was that all aerosol…" along with "A second assumption stated that all aerosol…" Thus, I opted to simply use the phrase, "One assumption is that all aerosol…" along with "Further, a second assumption is that all…" on page 7. Now the text on page 7 is consistent with that on page 11.

27) Suggestion on Page 8, line 2: I would sell that this thickness range is consistent with Figure 2. Added

28) Suggested a revision on Page 8, line 2-4: Reads awkwardly, please revise. I would suggest combining the statement (line 2-3) with the question (lines 2-4). Not enough information from you to understand "reads awkwardly".

29) Typo on Page 8, line 19: replace "diagnosed" with "derived" fixed

30) Suggestion on Pages 8 and 9: Your caption in Figure 4 point to Figure 5, which comes after it, which is not ideal. Because Figures 4 and 5 are so closely related, you may want to consider merging the two figures together as Figure 4a and 4b. This your annotations make more sense because you can directly see what is being emphasized without the need for additional explanation. Prior to Fig. 4 is a line that reads," There are also a few additional annotation symbols in Fig. 4 that will be discussed shortly." Because this text informs a reader that something will be discusses shortly, I removed the line from the caption of Fig. 4.

31) Suggested revision on Page 9, line 4: replace start with "Specifically, difference plots of brightness temperature values (Tbs) at 12.3. . ..". The original reads oddly with all the "of's". Addressed above.

32) Missing information on Page 9 lines 6 and 7: Although you are pointing to Miller et al. (2019), it would be more helpful if you stated what the critical value of vertical integrated water vapor is, otherwise your meaning comes off as rather vague. Is the content high? Low? Addressed above.

33) Removal on Page 8, lines 8 – 12: After "Although", the information is redundant and does not add anything to your manuscript. No "Although" appears on lines 8-12 on Page 8.

34) Removal on Page 8, Line 21-22: I would suggest removing sentence starting "With that" because it just repeats the same statement as the sentence coming before it. Instead skip right to the example. No "With that" appears on lines 21-22 on Page 8.

35) Add detail on Page 8, Line 4-5: As this is a new paragraph, I suggest adding the references to Figures 4 and 5 in parentheses after "channel difference image" and "GeoColor image". I don't see this; confused.

36) Typo on Page 10, line 6: should read as ". . .upper-left portion. . ." fixed

37) Suggested revision on Page 10, line 6: The white letter "A" is on Figure 4, yet you are talking about Figure 5. I would either mention that it is in Figure 4 or just say "denoted by the letter "A". Fixed, now reads, "is denoted by a letter "A".

38) Suggested merger on Page 11, line 12 and 13: Reads a little choppy, I would suggest merging the "Note" and that "That is" sentences. Disagree, the "That is" sentence serves to emphasize the previous sentence.

39) Revision or removal of sentence on Page 11, line 14-15. You clear illustrate that the TBs difference method for dust detection is not foolproof, but the latter part of the sentence reads a bit too vague because it is not clear as to what the "other components" refers to. Will you talk about them? Are you referencing other work? Some clarity here would help. You are referring to another part of my writing style. Sometimes I end a paragraph with a transitional sentence to advertise the next paragraph.

40) Page 12, line 15: Would be helpful to know what this critical integrated water vapor value is. Addressed above.

41) Typo on Page 12, line 20: "Since" should be "Because" Already got them.

42) Suggested modification on Page 13, line 13: Larger value can imply either larger positive or negative values. I would replacing "larger" with "positive" to remove any ambiguity. replaced

43) Suggestion on Page 13, line 23: Remove the "stated differently" phrase. You say the Tbs difference is 15C, yet -18C and -2C are 16C apart. Is this a typo or a rounding problem? When combined with the additional directions, it makes this statement more muddled. Removed "stated differently". Actually, the sentence reads, "Tbs decreased *about* 15˚ C from the Tb maximum to the north-northeast and southeast."

44) Suggested correction on Page 13, line 24: replace ". . .in values of Tbs of the. . ." with "in Tbs values of the. . .". To many prepositions here reads oddly. Addressed above

45) Typo on Page 14, line 6: "Since" should be "Because" fixed

46) Grammar error on Page 14, line 17: The colon use within the sentence makes no grammatical sense. Did you intend for a semi-colon? Please revise. ':' replaced with ';'

47) Suggested correction on Page 14, line 19: Need a better transition. You have the what "Values of TPW are shown in relation to various satellite fields". This sentence is grammatically correct, but it is a statement without the context. You give me a statement, but not a motivation on what you will do. "I am doing X to investigate Y." I endeavor to make the first sentence of a paragraph a so-called "topic sentence". A topic sentence serves to advertise the paragraph; as such, details are found in the body of the paragraph.

48) Minor correction on Page 16, line 5: "Largest" can be both positive or negative, it would be unambiguous if either "warmest" or "highest" were used instead. As seen in Fig. 8, there does not exist positive values of Tbs; no ambiguity exists.

49) Suggested edit on Page 16, line 15: Although you mention it in your caption, it would be worthwhile to also mention the channels being differenced are 10.35 and 12.3 microns. I use Tb(10.35 um)-Tb(12.3 um) many, many times in the paper. Further, as you pointed out, I mention it in the caption.

50) Suggested revision on Page 16, line 15-18: Sentence starting with "Not only" is quite wordy and hard to follow. I would suggest revising it into a more concise form such as "These figures show the NDR to be co-located with a, b, and c." Again, writing style. I'm using the "not only …, but also…" correlative conjunctions construct, which are coordinating conjunctions used in pairs.

51) Sentence too vague or out of place on Page 16, lines 19-20: The last sentence is terribly vague and just seems out of place. Which sensors? What is the significance? Writing style again, the last sentence is used as a transitional sentence to advertise the next paragraph.

52) Suggestion on Page 18, line 7: You mention that ALPW decreases from south to north and I can see this in your data. Perhaps it might be worth adding an annotation to Figures 12 to show exact where the 27.9 mm and 15.4 mm ALPW values are being estimated. Adding contours to Fig. 12 produces an ugly figure; hence they were left off the image.

53) Suggested replacement on Page 19, line 2. Replace "retrieval of values of ALPW" with "retrieval fo ALPW values" to remove excessive prepositions. Addressed above.

54) Possible logic error on Page 19, lines 2-3: The sentence has circular logic because "values of TPW are show in Fig. 15", but then later in the same sentence say it "was the time of the granule show in Fig. 15". Do you mean to reference a different figure? All is fine here. When one sees an image of a LEO with a time attached, a question arises about the time. Sometimes a time is relative to the LEO crossing a reference point in its orbit; at other times, the time refers to the granule. There are many granules in one orbit.

55) Suggestion on Figs 12, 13, and 14: You often cite 15N because it divides your pristine NDR and polluted SDR environments. I think it would be useful to consider annotating a line to each of these figures to show the 15N parallel for reference, especially for figure 14 where the coastline is muddied by the show Tb difference data. Fig. 2a provides the latitudes.

56) Overly wordy phrase on Page 20, line 6: "values of the difference between the temperature and dewpoint temperature". I would consider revising to be more concise. Disagree. I'm using a ";" to combine phrases.

57) Suggestion on Page 20, line 19: Instead of using the phrase "appeared earlier in the manuscript", I would consider adding "(Fig. 10)" after "images" on line 18 to make your wording more exact. Change made.

58) Typo on Page 23, line 7: should read as ". . .assimilation of aerosols. . ." Paragraph removed to shorten text as referenced above.

59) Awkward working on Page 23, line 7, ". . .assimilation of the coupled atmospheric component. . .". It is no clear as to what the atmosphere is coupled, I would suggest removing "coupled" to make the distinction between WCDA and SCDA more clear. Paragraph removed to shorten text as referenced above.

60) Typo on Page 23, line 14: should read as ". . .data of aerosols. . ." Paragraph removed to shorten text as referenced above.

61) Typo on Page 23, line 19: should read as ". . .a NWP model." Paragraph removed to shorten text as referenced above.

62) Logic error on Page 24, line 17: Once over the Atlantic, the SAL is by definition an EML. Section 5 is addressed above.

63) Typo on Page 24, line 36: should read as ". . .split-window difference. . ." fixed

64) Additional detail needed Page 25, line 8: It would be useful to know what this "some value" is. Addressed above

65) Typo on Page 25, line 20 and line: remove the space before " I don't see a double quote in that line.

66) Typo on Page 27, line 17: Remove the hyperlink associated with the doi Removed.

Kind Regards,

Louie

---

## Author Response (AR2)

Date: 04 January 2021

Purpose: Reply to AE

Uploaded Document Name: LGrasso_African_Dust_04_Jan_2021_reply_to_AE.pdf

Specific comments:

1. In the abstract (page 1, line 18). Should the "or" between "layer" and "TPW" be deleted?

Yes; deleted.

2. Page 13, Lines 19-21. I am confused by "cool Tbs associated with smoke may be confused with cool, elevated, land surfaces". Can you elaborate on the relevance of this statement since the bulk of the analysis in the manuscript occurs over the ocean?

I agree that there appears to be a strained connection between a discussion of smoke over land and a discussion of smoke over the eastern Atlantic Ocean. My goal is to provide a reader with a complete description of satellite detection of smoke, as opposed to an ocean specific discussion. As such, I did make a small modification of the text as follows: "Based on previous satellite observations, Hillger and Ellrod (2003) have shown that smoke layers were undetected in values of infrared channel differences. In an attempt for this manuscript to provide a more complete background of satellite detection of smoke, Hillger and Ellrod (2003) also showed that if a layer of smoke is optically thick enough in infrared bands, Tbs of smoke will appear cool and may be confused with cool, elevated, land surfaces."

3. Page 22, Line 7-8. "However, both methods relied on measurements of reflected solar energy...". CALIOP is an active sensor that relies on transmitted laser energy to detect the vertical distribution of aerosol and cloud layers using backscattered signal. CALIOP actually has better performance during the night because there isn't any solar contribution to the measured signal. Please consider revising/rephrasing this section of the paragraph.

Yes, my mistake. I changed the text from

"As discussed above, GeoColor imagery and the CALIOP instrument on CALIPSO detected dust in the SDR. However, both methods relied on measurements of reflected solar energy; as a result, dust will go undetected after sunset in the SDR. However, dust in the NDR was not only detected, but also tracked after sunset."

to

"As discussed above, GeoColor imagery detected dust in the SDR. However, GeoColor imagery relied on measurements of reflected solar energy; as a result, dust will go undetected after sunset in the SDR. However, dust in the NDR was not only detected, but also tracked after sunset."